# HATRIC-based identification of receptors for orphan ligands

Nadine Sobotzki[1,6], Michael A. Schafroth[2], Alina Rudnicka[3,7], Anika Koetemann[1], Florian Marty[4,8], Sandra Goetze[1], Yohei Yamauchi[5], Erick M. Carreira[2] & Bernd Wollscheid[1]

Cellular responses depend on the interactions of extracellular ligands, such as nutrients, growth factors, or drugs, with specific cell-surface receptors. The sensitivity of these interactions to non-physiological conditions, however, makes them challenging to study using in vitro assays. Here we present HATRIC-based ligand receptor capture (HATRIC-LRC), a chemoproteomic technology that successfully identifies target receptors for orphan ligands on living cells ranging from small molecules to intact viruses. HATRIC-LRC combines a click chemistry-based, protein-centric workflow with a water-soluble catalyst to capture ligand-receptor interactions at physiological pH from as few as 1 million cells. We show HATRIC-LRC utility for general antibody target validation within the native nanoscale organization of the surfaceome, as well as receptor identification for a small molecule ligand. HATRIC-LRC further enables the identification of complex extracellular interactomes, such as the host receptor panel for influenza A virus (IAV), the causative agent of the common flu.

[1] Department of Health Sciences and Technology & Institute of Molecular Systems Biology & BioMedical Proteomics Platform (BMPP), ETH Zurich, Zurich, Switzerland. [2] Department of Chemistry and Applied Biosciences, Laboratory of Organic Chemistry, ETH Zürich, Vladimir-Prelog-Weg 3, Zürich 8093, Switzerland. [3] University of Zurich, Institute of Molecular Life Sciences, Winterthurerstrasse 190, Zurich CH-8057, Switzerland. [4] Dualsystems Biotech AG, Wagistrasse 12, Schlieren 8952, Switzerland. [5] School of Cellular and Molecular Medicine, University of Bristol, Biomedical Sciences Building, University Walk, Bristol BS8 1TD, UK. [6] Present address: Merck Ventures B. V., Gustav Mahlerplein 102, 1082MA Amsterdam, The Netherlands. [7] Present address: School of Cellular and Molecular Medicine, University of Bristol, Biomedical Sciences Building, University Walk, Bristol BS8 1TD, UK. [8] Present address: Biognosys AG, Wagistrasse 21, Schlieren 8952, Switzerland. Correspondence and requests for materials should be addressed to B.W. (email: wbernd@ethz.ch)

Physiological ligand-receptor interactions are typically of low affinity and occur under native conditions, making them difficult to study in vitro[1]. Thus, the receptors for many ligands have not been identified. Ligand-based receptor capture (LRC) technology partly overcame these difficulties and enabled the identification of ligands for orphan N-glycoprotein-receptors using the tri-functional reagent TRICEPS[2,3] and modifications thereof using a cross-linker containing an aldehyde-reactive aminooxy group, a sulfhydryl, and a biotin group (ASB)[4]. Application of TRICEPS-LRC and ASB in different biological systems revealed the need to redesign the first-generation technologies: TRICEPS-LRC was intentionally designed to enable the identification of ligand-bound receptors solely based on formerly N-glycosylated peptides. O-glycosylated receptors and N-glycosylated receptors, whose deamidated peptides were not detectable by mass spectrometry, were eventually missed by this strategy. However, this peptide-based strategy benefitted from the ability to filter for deamidated receptor peptides as indicators of direct TRICEPS-crosslinking and ligand-binding. In contrast, in ASB, tryptic digestion is performed directly on Streptavidin beads, which enables protein-level affinity purification, enabling, in principle, the identification of receptors through non-glycopeptides. However, direct digestion of proteins bound to Streptavidin beads leads to major contaminations with streptavidin peptides, impairing identification and label-free quantification of receptor peptides. Furthermore, ASB requires performing a two-step reaction in order to couple the ligand to the cross-linker, and biotin transfer from ligand to receptor is mediated by reduction of a disulfide bond, making its application sensitive to reductive environments. Additionally, similar to first-generation TRICEPS-LRC, ASB requires high amounts of starting material (50 million cells or 5–7 150 mm plates) and captures ligand–receptor interactions in presence of a catalyst for oxime formation at non-physiological pH 8, compared to pH 6.5 for TRICEPS-LRC. The pH of the microenvironment directly influences the affinity between a ligand and its receptor, exemplified by ligands that are internalized upon receptor binding: The affinity for the receptor is high at pH 7.4 on the surface of living cells, but decreases upon acidification (pH 6.5) in the endosome, leading to release of the ligand from the receptor. A prime example of this is folate, which has an affinity for folate receptor alpha (FOLR1) that is 2000 times lower at pH 6.5 than at pH 7.4[5]. Consequently, the folate receptor has not been detected by TRICEPS-LRC in the past, highlighting the need for a next-generation LRC suited for receptor deorphanization at physiological pH. Here we develop a new LRC technology with catalyst-enhanced cell surface labeling and protein-level affinity purification that enables the capture of ligand-receptor interactions at pH 7.4 from as few as 1 million cells.

## Results

### Development of a HATRIC-based LRC workflow

The next-generation LRC technology is based on a new tri-functional compound, HATRIC, with an acetone-protected hydrazone, an N-hydroxysuccinimid (NHS), and an azide (Fig. 1a, Supplementary Note 1). First, the ligand is linked through a primary amine to the NHS-moiety of HATRIC (Fig. 1b). Second, living cells are mildly oxidized with sodium-meta-periodate to generate aldehydes from cell surface carbohydrates. During the whole experiment, cells are kept on ice to prevent any receptor-mediated internalization events. Third, the HATRIC-ligand conjugate is added to the cells in the presence of catalyst 5-methoxyanthranilic acid (5-MA) and receptor-capture is performed at pH 7.4. The ligand enhances local HATRIC reactivity in the vicinity of the target receptor or receptors, and receptor

aldehydes react with the acetone-derived hydrazone of HATRIC. In the control, the HATRIC-conjugated ligand is applied to the cells in the presence of an excess unmodified ligand. Here, the out-competed ligand–HATRIC conjugate reacts randomly with cell surface glycoproteins. As alternative controls, HATRIC can be quenched with glycine (negative control) or a ligand with known target receptors can be employed as a positive control (not depicted in Figure 1). After the reaction, cells are lysed, and azide-tagged surface proteins are affinity-purified by linking azide-tagged proteins to alkyne agarose using copper-based click chemistry[6]. Trypsin-mediated proteolysis of bead-bound proteins releases the unglycosylated peptides. These peptides are analyzed with high-accuracy mass spectrometry using data-dependent acquisition and filtered for known and predicted cell surface proteins. The quantitative comparison to the competitive control reaction reveals specific enrichment of target cell surface receptors for the ligand. The novel workflow renders HATRIC-LRC independent of the PNGase F deglycosylation reaction, ultimately enabling a more robust relative quantification of cell surface receptors than is possible with first-generation LRC.

To enable HATRIC-LRC under physiological conditions, it was necessary to accelerate the reaction of hydrazines with aldehydes, which is slow at neutral pH[7]. Aniline has been exploited to catalyze similar reactions efficiently[8]; however, the cytotoxicity at the required concentration limits use with living cells[9]. Aniline-derived water-soluble catalysts have been described that substantially improve catalysis of hydrazone formation, but none had been tested in biological systems[10]. Evaluation of a number of aniline derivatives regarding their solubility, cytotoxicity and capability to enhance hydrazone formation between aldehydes on cell surface proteins and the HATRIC-hydrazide on living cells led to the identification of 5-methoxyanthranilic acid (5-MA, Fig. 1c, Supplementary Fig. 1). 5-MA catalyzed hydrazone formation at a non-toxic concentration at pH 7.4 more efficiently than 2-amino-4,5-dimethoxy benzoic acid (ADA) (Fig. 1c). Additionally, replacing the original Trifluoroacetyl-protection group of TRICEPS by an acetone-derived protection group in HATRIC enabled higher yield of hydrazone formation on living cells. Last, we confirmed that, under the chosen conditions, HATRIC does not penetrate cells avoiding contamination with intracellular proteins (Supplementary Fig. 2).

### Validation of HATRIC-based LRC

We validated HATRIC-LRC demonstrating capture of epidermal growth factor receptor (EGFR) using epidermal growth factor (EGF) as a ligand in an experiment with living H-358 cells (Fig. 2a). When initially quantifying all identified proteins across samples, we found 9 proteins significantly enriched in the EGF-captured samples, but only three of them were cell surface proteins, and EGF as ligand dropped below significance level. Statistical scoring of protein candidates is based on the number of peptides identified per proteins, which leads to bias towards larger proteins or proteins whose peptides are easily detectable in MS (e.g., 19 features were quantified and scored statistically for EGFR, whereas only 1 peptide was quantified and scored for EGF). In order to overcome this bias, we used a filter for known and predicted cell surface proteins prior to statistical scoring to rescue receptor candidates where most peptides are hardly detectable via MS (e.g., due to decreased solubility) (Supplementary Fig. 3; Supplementary Data 1–3). Applying this filter prior to statistical analysis, we correctly found EGF significantly enriched and identified five other EGF receptor candidates that have not been described before (Supplementary Table 1, Supplementary Data 4 and 5), namely monocarboxylate transporter 4 (SLC16A3), filamin-A (FLNA), peroxisomal 3-ketoacyl-CoA thiolase (ACAA1),

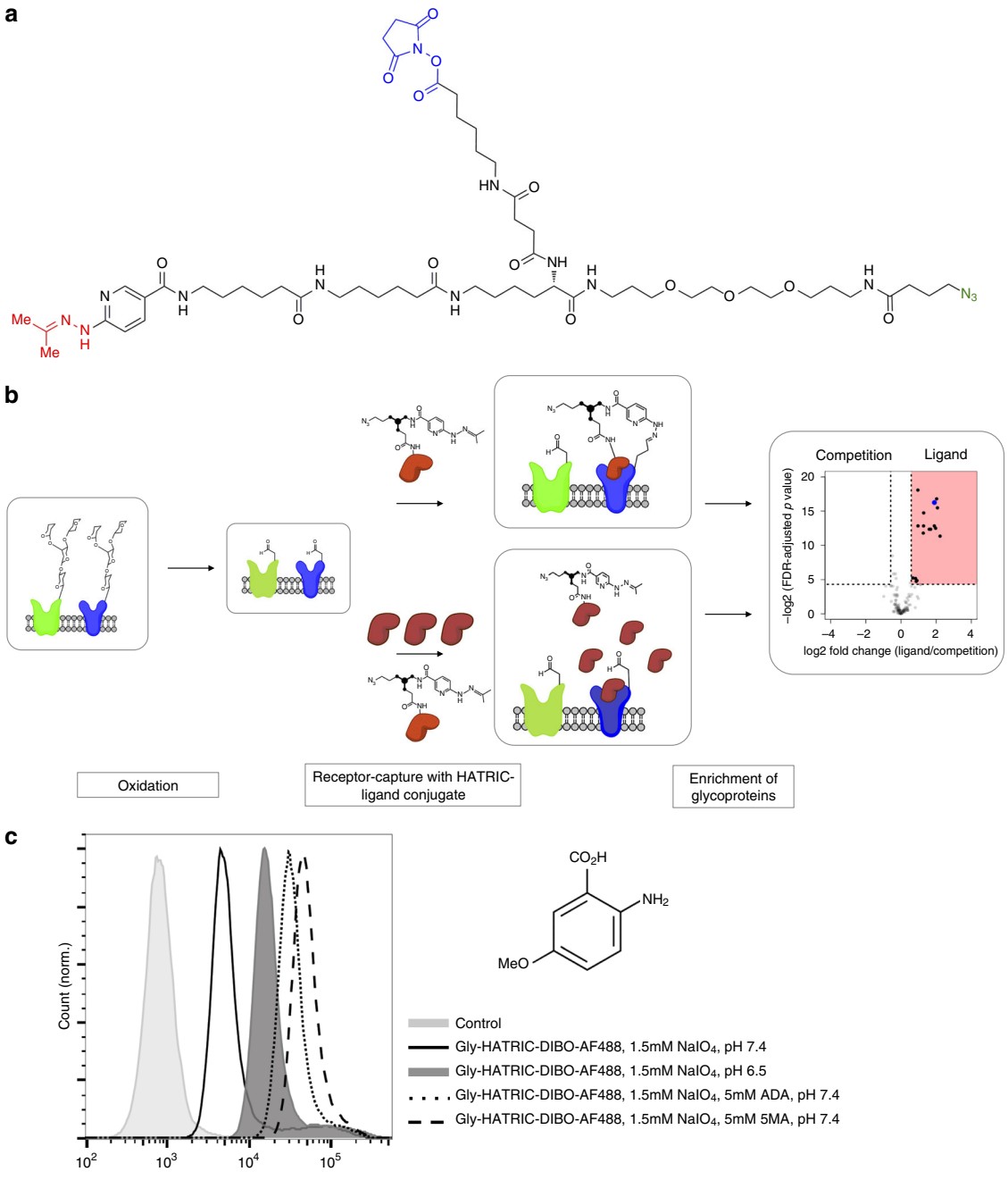

**Fig. 1** HATRIC-LRC enables ligand-based receptor capture. **a** Structure of hydrazone (highlighted in magenta) and azide (highlighted in turquoise) containing tri-functional compound HATRIC (NHS ester highlighted in blue). Mw = 1171.4 g mol$^{-1}$ (synthesis described in Supplementary Note 1). **b** Workflow of HATRIC-LRC for identification of target receptors of ligands on living cells. First, living cells are mildly oxidized with 1.5 mM NaIO4. HATRIC, conjugated to the ligand of interest, is added to living cells. The ligand selectively directs HATRIC to its glycoprotein target receptor, where HATRIC reacts to generate azide-tagged cell-surface glycoproteins catalyzed by 5-MA. In order to identify target receptors of orphan ligands, a dual track experimental setup is employed. In the control, the HATRIC-conjugated ligand is applied to the cells in the presence of an excess unmodified ligand. Alternatively, HATRIC can be quenched with glycine for a negative control or a ligand with known target receptors can be employed as a positive control (not depicted in figure). After lysis and affinity purification of azide-tagged proteins with unbound proteins removed by harsh washing, peptides are proteolyzed with trypsin. Peptides are identified with high-accuracy mass spectrometry in a data-dependent acquisition mode followed by quantitative comparison of peptide fractions? from experiment and control to reveal specific enrichment of candidate cell surface receptors. Target receptors are defined as proteins that have a fold change of >1.5 compared to the control as well as an FDR-adjusted $p$ value (Benjamini–Hochberg method) equal to or smaller than 0.05, corresponding to a target receptor window in the volcano plot that is framed by dotted lines and highlighted in red. **c** Flow cytometry traces of U-2932 cells incubated with HATRIC conjugated to dibenzocyclooctyne-Alexa Fluor 488 (DIBO-AF488) at pH 6.5 or pH 7.4 in the presence or absence of organocatalyst 5-methoxyanthranilic acid (5-MA) (structure shown, Mw = 167.16 g mol$^{-1}$) or 2-amino-4,5-dimethoxy benzoic acid (ADA). HATRIC was quenched with glycine (Gly−) to avoid potential reaction of HATRIC's NHS-ester with amino groups at the cell surface. Shift to the right indicates more efficient labeling with HATRIC-DIBO-AF488

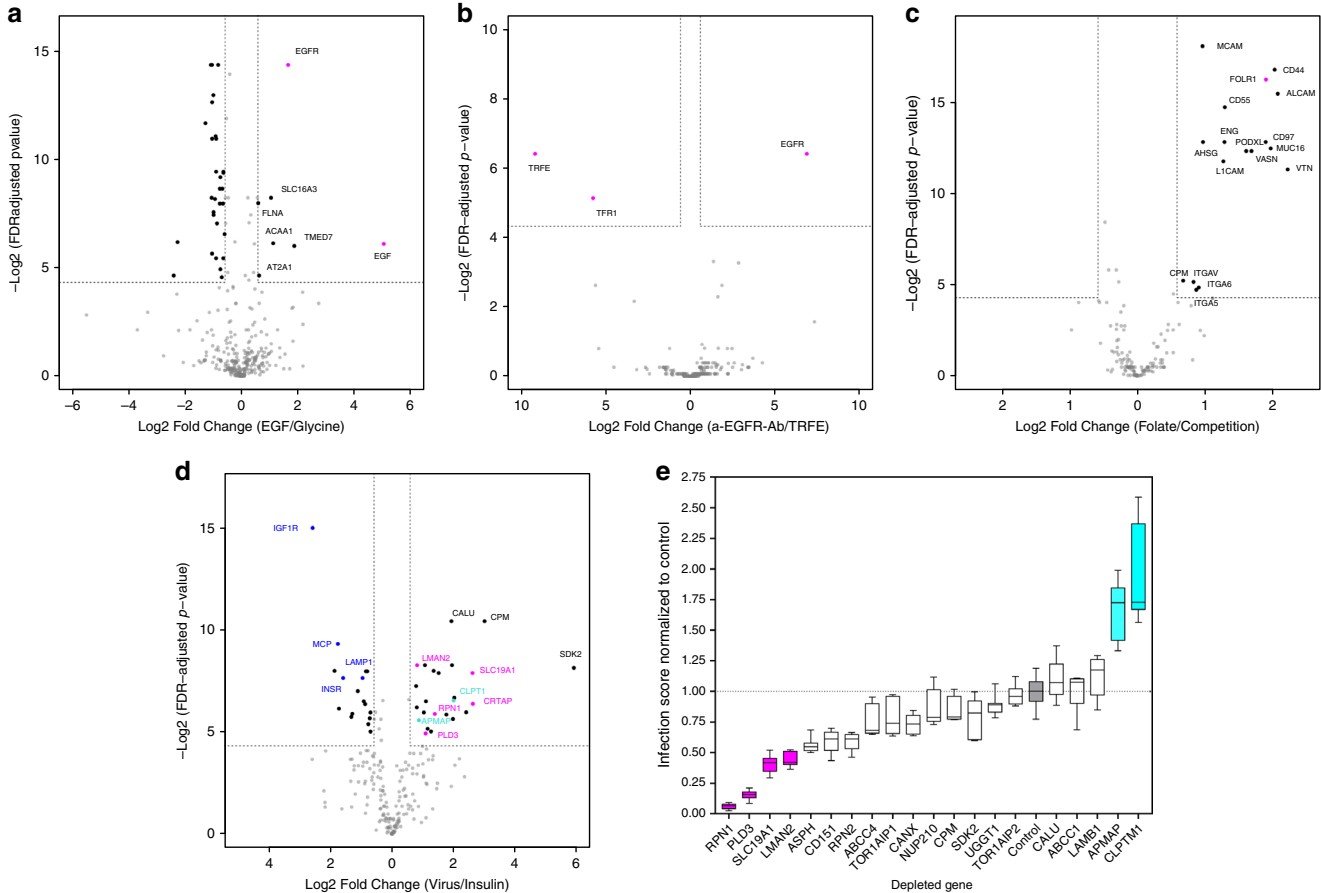

**Fig. 2** HATRIC-LRC identifies target receptors for ligands ranging from small molecules to intact viruses. All results of HATRIC-LRC experiments are presented in volcano plots, where fold changes of proteins are shown with their respective log-transformed, fold changes are also log transformed false-discovery rate (FDR)-adjusted p values. Target receptors are defined as proteins that have a fold change of greater than 1.5 compared to the control as well as a p value equal to or smaller than 0.05 (Benjamini–Hochberg method), corresponding to a target receptor window in the volcano plot that is framed by dotted lines. All experiments were performed in triplicates per condition, except for the H3N2, where quadruplicates were produced. **a** HATRIC-LRC with EGF on 20 million H-358 cells. In the negative control, HATRIC was quenched with glycine to map the off-target reaction of HATRIC on the same cell line. The ligand and the known target receptor are highlighted in magenta. **b** HATRIC-LRC experiments with EGF and TFRE were performed on 1 million MDA-MB-231 cells. In this experiment, two ligands with known receptors served as controls for each other to benchmark the ability to perform HATRIC-LRC with as little as 1 million cells. The ligands and known target receptors are highlighted in magenta. **c** Folate-based HATRIC-LRC was performed on 20 million folate-starved HeLa Kyoto cells per replicate. In the control, six-fold excess of free folate was used to compete with binding of folate-HATRIC. The target receptor FOLR1 is highlighted in magenta. **d** IAV-based HATRIC-LRC was performed on 20 million A549 lung adenocarcinoma cells per replicate. In the positive control, insulin was used as ligand, and insulin receptors were correctly identified. In the IAV-target receptor window, magenta-colored red dots highlight receptors that showed an inhibitory effect on IAV cell entry whereas turquoise dots blue highlight receptors that facilitated IAV cell entry in a siRNA-based knockdown experiment (Fig. 2e). **e** Effect of siRNA-mediated depletion of candidate receptors on IAV infection of A549 cells. Experiments were conducted in triplicates. Infection scores from siRNA-treated samples were normalized to control samples transfected with non-targeting siRNA (shown in gray). The data are presented as boxplots with whiskers from minimum to maximum values. Boxes extend from the 25th to 75th percentiles. The line in the middle of the boxes depicts the median. The dotted line on the plot shows the median of control group? (normalized to 1). Magenta and turquoise Red and green boxes highlight receptors that showed an inhibitory or facilitative effect on IAV cell entry (Magenta Red: infection score decreased by >50%, turquoise green: infection score increased by >50% upon gene depletion)

transmembrane emp24 domain-containing protein 7 (TMED7) and sarcoplasmic/endoplasmic reticulum calcium ATPase 1 (AT2A1) (Supplementary Table 1). Reports of direct interactions between these proteins and EGF are not available, but it was shown before that SLC16A3 co-locates with CD147 in breast cancer cells[11], which in turn is associated with EGFR in similar lipid domains[12], suggesting that SLC16A3 resides in the neighborhood of EGFR at the cell surface[13].

**Identifying cell surface receptors from 1 million cells.** As HATRIC-LRC is based on protein-level purification, more than

one peptide is commonly identified per protein, such as exemplified by EGFR (Supplementary Fig. 4). Therefore, we investigated the HATRIC-LRC detection limit with respect to the amount of starting material needed for successful receptor identification. From as little as one million MDA-MB-231 cells per replicate, we were able to unambiguously identify EGFR as the receptor for HATRIC-coupled anti-EGFR antibody and transferrin receptor protein 1 (TFR1) as the receptor for HATRIC-coupled Holo-transferrin (TRFE) (Fig. 2b, Supplementary Data 6 and 7), which was not possible with TRICEPS-LRC (Supplementary Figures 5 and 6, Supplementary Data 8

**Table. 1 Folate-receptor candidates identified using folate-based HATRIC-LRC**

| Protein name | Gene name | Uniprot accession | log2-transformed fold change | FDR-adjusted *p* value |
|---|---|---|---|---|
| Vitronectin | VTN | P04004 | 2.22 | 0.00 |
| CD166 antigen | ALCAM | Q13740 | 2.08 | 0.00 |
| CD44 antigen | CD44 | P16070 | 2.03 | 0.00 |
| Mucin-16 | MUC16 | Q8WXI7 | 1.97 | 0.00 |
| Folate receptor alpha | FOLR1 | P15328 | 1.90 | 0.00 |
| CD97 antigen | CD97 | P48960 | 1.90 | 0.00 |
| Vasorin | VASN | Q6EMK4 | 1.69 | 0.00 |
| Podocalyxin | PODXL | O00592 | 1.61 | 0.00 |
| Complement decay-accelerating factor | CD55 | P08174 | 1.29 | 0.00 |
| Endoglin (CD105) | ENG | P17813 | 1.29 | 0.00 |
| Neural cell adhesion molecule L1 | L1CAM | P32004 | 1.27 | 0.00 |
| Alpha-2-HS-glycoprotein | AHSG | P02765 | 0.97 | 0.00 |
| Cell surface glycoprotein MUC18 | MCAM | P43121 | 0.96 | 0.00 |
| Integrin alpha-6 (CD49F) | ITGA6 | P23229 | 0.91 | 0.03 |
| Integrin alpha-5 (CD49E) | ITGA5 | P08648 | 0.87 | 0.04 |
| Integrin alpha-V | ITGAV | P06756 | 0.83 | 0.03 |
| Carboxypeptidase M | CPM | P14384 | 0.68 | 0.03 |

and 9). Where possible, we recommend the usage of 5–20 million cells in order to detect low copy number receptors based on a given sensitivity of the MS-instrument used for analysis.

**Identification of receptors for small molecule ligands**. Next, we investigated if HATRIC-LRC technology could be utilized to identify receptors for small-molecule ligands. Previous studies have focused on high-affinity interactions of small molecules with cytoplasmic proteins, resulting in underrepresentation of cell-surface proteins in these studies[14]. Cell surface interactions of small molecules, however, are clearly important, especially for their uptake into cells[15]. As a proof-of-concept, we investigated interactions of folate, which is known to interact with the cell-surface protein folate receptor alpha (FOLR1). We synthesized a folate-HATRIC conjugate (Supplementary Note 2) guided by available structural data on the folate-FOLR1 complex[16–18]. We incubated the folate-HATRIC conjugate with 20 million HeLa Kyoto cells at pH 7.4. In the control, we competed with a six-fold excess of unmodified folate. We detected interactions with FOLR1 and with a small set of further receptor candidates (Fig. 2c, Table 1; Supplementary Data 10 and 11). None of these receptors were previously described to directly interact with folate and we did not identify any other known folate receptors. We speculate that other folate receptors (e.g., FOLR2) were not identified as their affinity towards folate is lower than the affinity of FOLR1, i.e., FOLR2 has a two-fold reduced affinity for folate compared to FOLR1, or because they are not expressed in HeLa Kyoto cells[19]. Related approaches studied methotrexate-based labeling of FOLR1, but did not investigate whether the compound also binds to other proteins[18]. This experiment shows that HATRIC-LRC can be used as a screening method to identify relevant receptor candidates for small-molecules that can be functionally validated in subsequent experiments.

**Identification of receptor candidates for influenza virus**. We next explored whether HATRIC-LRC could be used to identify entry facilitators for viruses. Currently, the contribution of cell-surface proteins as entry facilitators remains poorly understood. Viruses such as Influenza A virus (IAV) are known to have multiple pathways of entry that depend on host cell type, availability of cell-surface receptors, and state of cell polarization[20]. To infect cells, IAV-particles first attach to cell surface sialyl-oligo-saccharides, which are recognized by the IAV surface glycopro-tein hemagglutinin (HA)[21]. This interaction results in virus attachment to the host cell but can not trigger cell entry, as sialic acids do not possess signaling capacity[22]. After endocytosis, the virus travels through the endocytic pathway. In late endosomes, the low pH initiates viral fusion, followed by completion of uncoating, penetration, and infection. Genome-wide RNAi studies have revealed a number of protein-based receptor candidates for IAV, but their precise roles in infection remain unclear[23]. We used HATRIC-LRC to shed light on the complex interactions between IAV and its host cells. Human IAV H3N2 (strain X-31) was coupled to HATRIC and it was demonstrated that, although coupling reduced IAV endocytosis, the particles used similar endocytic pathways to the wild-type virus (Supplementary Figures 7 and 8). We conducted H3N2-based HATRIC-LRC on 20 million human lung adenocarcinoma (A549) cells and compared to the control ligand insulin. We identified 24 virus-interacting candidates (Fig. 2d, Supplementary Table 2, Supplementary Data 12 and 13).

**Verifying influenza entry facilitator receptor candidates**. To determine whether candidate receptors impact IAV entry, we depleted A549 cells of 21 of these proteins using short interfering RNA (siRNA) and analyzed infection efficiency. siRNA-mediated depletion of >70% was confirmed by real-time RT-real time vs RT PCR in 20 genes. We excluded cartilage-associated protein (CRTAP) from further analysis as siRNA treatment failed to deplete it (Supplementary Fig. 9, Supplementary Data 14). Depletion of four proteins, phospholipase D3 (PLD3), ribophorin I (RPN1), folate transporter 1 (SLC19A1) and vesicular integral-membrane protein VIP36 (LMAN2) reduced IAV infection by more than 50% relative to cells treated with control siRNA (Fig. 2e, Supplementary Data 15). Depletion of RPN1 reduced IAV infection by over 94%. RPN1 is a subunit of the proteasome regulatory particle, and thus might have effects on cells that are not directly related to viral entry. However, its presence at the cell membrane indicates that RPN1 may serve as an IAV entry facilitator that could be a target for pharmaceutical intervention. In contrast, depletion of two other identified proteins, adipocyte plasma membrane-associated protein (APMAP) and cleft lip and palate transmembrane protein 1 (CLPTM1), led to an increase of infection by over 70% (Fig. 2e), indicating that these factors restrict viral entry under physiological conditions. Of the receptor candidates identified using HATRIC-LCR, none have been implicated previously in mediating H3N2 infection. However, it has been shown that related phospholipase γ1 (PLC-γ1) signaling is activated by IAV H1N1 and mediates efficient viral entry in

human epithelial cells[24]. Of the 5 independent genome-wide siRNA screens on IAV that were published, three have been validated[25–27]. Of the 129, 168 and 219 genes, respectively, that were validated as hits from these three screens, 34 were shared in two or more. Only 3 genes (ARCN1, ATP6AP1, and COPG) were shared among all three. A comparison of our top seven decreaser genes (RPN1, PLD3, SLC19A1, LMAN2, ASPH, CD151, RPN2) with the 34 genes revealed mild functional overlap[28]. We also had 4 strong hits (i.e., increased or decreased infection by more than 70%) out of 20 validated genes—a hit rate of 20%—which is considerably higher compared to the genome-wide screens (~1%). The identification of these potential IAV entry facilitators suggests potential for new lines of influenza research and therapy.

## Discussion

Given the experimental setup, the candidates identified from HATRIC-LRC experiments can generally result from the following four scenarios: (1) there is a direct interaction of the ligand with the target receptor; (2) the protein is in close proximity of the target receptor ("neighborhood protein"); (3) the protein is upregulated in response to treatment with the ligand and is overrepresented in the background binding of HATRIC (e.g., we use approximately 8 times more EGF than is used for stimulation experiments) or (4) the identified candidate is a false positive.

A single HATRIC-LRC experiment does not allow us to delineate which type of interaction is taking place, but the validation experiments and the cited data clearly underline the relevance of the identified proteins. The analysis pipeline was optimized to allow identification and ranking of receptor candidates. However, the resulting data have to be analyzed carefully, and more stringent receptor spaces can be defined based on the identification of positive control receptors or the ligand (e.g., EGF). Identified candidates need validation in tailor-made follow-up experiments, such as siRNA-based approaches.

We demonstrated that HATRIC-LRC enables ligand-receptor identification from as few as 1 million cells at physiological pH through new chemistry combining HATRIC, a water-soluble catalyst, and click chemistry-based protein-level affinity purification in a competition-based workflow. Even though HATRIC-LRC is a screening technology that leads to candidate receptors, including potentially false positive receptor candidates, which need to further validated, its ability to detect biologically meaningful ligand-receptor interactions remains unmatched. The power of HATRIC-LRC to detect functionally relevant cell surface interactions was demonstrated using ligands ranging from small molecules to intact influenza A virus particles. The IAV application showed that HATRIC-LRC is able to identify binding partners of multivalent virus particles on the cell surface that are potentially organized in a viral synapse. In future applications, HATRIC chemistry should enable the analysis of post-translational modifications of signaling competent receptors and, given the drastically increased sensitivity of the HATRIC technology, applications using rare biological or clinical specimens, opening up new avenues for biomedical research.

## Methods

**Chemicals**. All chemicals and cell culture reagents were from Sigma-Aldrich unless stated otherwise.

**Synthesis of HATRIC**. The trifunctional reagent HATRIC, containing an acetone-protected hydrazone and an azide affinity tag, was synthesized from Fmoc-Lys (Boc)-OH in seven steps using standard peptide coupling conditions.

**Synthesis of folate-HATRIC**. We synthesized a Boc-protected amine-functionalized folate derivative by peptide coupling of a glutamic acid derivative with pteroic acid. After deprotection, the resulting amine was conjugated to HATRIC, resulting in a folate-HATRIC conjugate (Supplementary Note 2). Structural data of the

folate-FOLR1 complex as well as previous modification of folate at this site supported the choice for the conjugation site[16–18].

**Mammalian cell culture**. Cells were grown at 37 °C and 5% ambient $CO_2$. HeLa Kyoto (ATCC), MDA-MB-231 (ATCC), and A549 cells (ATCC) were grown to ~80% confluence in 140 × 20 mm dishes (Nunclon™ Delta Airvent, Thermo Fisher) with Dulbecco's Modified Eagle's Medium, high glucose, GlutaMAX™ supplement (DMEM, Thermo Scientific), 10% fetal bovine serum (FBS), and 1% penicillin-streptomycin. H-358 (ATCC CRL-5807) and U-2932 (ATCC) cell lines were cultured in RPMI 1640 with GlutaMAX™, 1% penicillin-streptomycin, and 10% FBS. Prior to HATRIC-LRC experiments with folate, HeLa cells were starved for 48 h in folate-free RPMI (Gibco).

**Catalyst cytotoxicity assays**. MDA-MB 231 cells (20.000 cells per well in a 96-well plate) were treated with the indicated concentrations of catalyst in DMEM (pH adjusted to 7.4, 1% Pen/Strep) for 1.5 h at 37 °C. Supernatant was replaced by 100ul DMEM with 10% alamarBlue™ reagent (ThermoScientific) and incubated for 5 h at 37 °C in the dark. Assay was read out by a fluoreader (Ex: 545 nm, Em: 590 nm, automatic gain).

**Confocal microscopy imaging**. HATRIC was pre-coupled to equimolar amine-Cy3 (Lumiprobe) in 25 mM HEPES (pH 8.2) for 1.5 h at RT and 300 rpm in the dark. MDA-MB-231 cells cultured on coverslips were oxidized with 1 ml of 1.5 mM sodium periodate in PBS, pH 6.5 for 15 min and labeled with 6 μM HATRIC-Cy3 or amine-Cy3 (Control w/o HATRIC) in 1 ml PBS with 5 mM 5-MA (pH 7.4) for 1.5 h at 4 °C shaking in the dark. As a cell surface marker, cells were labeled with 0.5 ml 1 mM sulfo-NHS-Cy5 (Lumiprobe). Nuclei were stained with 0.5 ml 1 μg ml$^{-1}$ Hoechst (Molecular Probes H1399) for 10 min at 4 °C. Cells were fixed with 4% paraformaldehyde for 10 min at RT, mounted with anti-fade mounting medium (Molecular Probes Prolong Gold Antifade reagent P36934) and analyzed by confocal microscopy (Leica TCS SP2). For a permeabilized control, cells were first stained with sulfo-NHS-Cy5 and fixed, and then permeabilized with 0.1% Triton X-100 for 10 min at RT, before oxidation and labeling with HATRIC-Cy3.

**Visualization of cell surface azidylation with HATRIC**. Amine-reactive groups of HATRIC were quenched with 10-fold excess glycine, and the HATRIC azide was reacted with 50 mM Click-IT Alexa Fluor 488 DIBO Alkyne (DIBO-AF488, Thermo Fisher Scientific) in a copper-free click chemistry reaction in 25 mM HEPES, pH 8.2 for 60 min. Per replicate, $1 \times 10^6$ U-2932 cells were oxidized with 1.5 mM $NaIO_4$ for 15 min at 4 °C in PBS (pH 6.5) with slow rotation in the dark or left untreated. Cells were washed twice with phosphate-buffered saline (PBS, pH 7.4) to remove residual $NaIO_4$. Cells were labeled with 75 μM glycine-quenched HATRIC-DIBO-AF488 conjugates for 60 min at 4 °C with slow rotation in the presence or absence of 5 mM 5-MA or 5 mM ADA. The pH was adjusted to 6.5 with $H_3PO_4$ or to 7.4 with NaOH prior to addition to cells. Unstained cells were washed twice with FACS buffer (PBS containing 1% FBS). Stained cells were washed four times with FACS buffer and resuspended in 500 μl FACS buffer. Cells were analyzed on an Accuri C6 Flow Cytometer (BD Biosciences). The data were analyzed with FlowJo v.10.

**HATRIC-based ligand-receptor capture**. All experiments were performed in triplicates per experiment, except for the H3N2, which was produced in quadruplicates. Each experiment was reproduced at least once in the laboratory. HATRIC-LRC was performed in seven steps, as follows: (1) Coupling of trifunctional compound: Ligand (100 μg insulin, 100 μg EGF, 100 μg anti-EGFR antibody (E2156-200UL, Sigma-Aldrich), 100 μg transferrin or or $5 \times 10^8$ plaque-forming units of X-31) was reacted with 70 μg HATRIC (100 mM stock solution in DMSO) for 1.5 h with slow rotation at 22 °C in 50 μl 25 mM HEPES (pH 8.2). For HATRIC-small molecule conjugates, pre-coupling was not necessary. (2) Collection of cells and oxidation: Per replicate, the indicated number of cells were collected by gentle scraping (adherent cell lines) or centrifugation (suspension cell lines) and collected at 4 °C. Cells were resuspended in PBS (pH 6.5) and oxidized for 15 min with 1.5 mM $NaIO_4$ at slow rotation at 4 °C in the dark. Cells were washed once with PBS (pH 6.5) and resuspended in 10 ml PBS containing 5 mM 5-MA (pH 7.4). (3) Receptor capture with HATRIC: HATRIC-ligand solution (50 μl) was added to a final concentration of 6 μM HATRIC. For folate-HATRIC-LRC, a six-fold excess free compound was added to control reactions. Cells were incubated for 90 min at slow rotation and 4 °C. Cells were pelleted, washed twice with PBS (pH 7.4) to remove unbound HATRIC, and lysed with 8 M Urea, 0.1% RapiGest SF (Waters) containing protease inhibitors (cOmplete, Roche), pH 8. (4) Cell lysis: Cells were lysed using three 20 s sonication pulses in a VialTweeter. Debris was pelleted by centrifugation at 16,000 × $g$ for 10 min (4 °C), and supernatants placed on ice. (5) Click chemistry-based enrichment: Per replicate, 200 μl alkyne agarose (Jena Bioscience) was washed three times with 1.8 ml MilliQ (MQ) water. The lysates were added to the beads, and 1 ml click chemistry buffer was added with final concentrations of 1 mM $CuSO_4$, 6.25 mM THPTA, and 10 mM sodium ascorbate. The copper-catalyzed azide-alkyne cycloaddition reaction was conducted for 18 h with slow rotation at room temperature. (6) Stringent washing of resin: Agarose beads were pelleted by centrifugation for 4 min at 300 × $g$, supernatant was

removed, and beads were washed twice with 1.8 ml MQ water. Cells were resuspended in 1 ml sodium dodecyl sulfate (SDS) wash buffer (1% SDS, 100 mM Tris, 250 mM NaCl, 5 mM EDTA, pH 8), and bead-bound proteins were reduced with 5 mM Tris(2-carboxyethyl)phosphine (TCEP) for 15 min at 55 °C and 15 min at room temperature. Beads were pelleted at $300 \times g$ for 4 min to remove supernatant. Bead-bound proteins were alkylated with 40 mM iodoacetamide for 30 min at room temperature in the dark. Beads were transferred to MoBiCol classic 35-μm filters (MoBiTec) and washed with 10 ml SDS wash buffer, 10 ml 8 M urea/100 mM Tris, 5 ml 5 M NaCl, 1.5 ml 80% isopropanol, 5 ml 100 mM NaHCO₃ (pH 11), 5 ml 50 mM ammonium bicarbonate (60 °C), and 5 ml 20% acetonitrile. Then beads were transferred to fresh MoBiCols. (7) On-bead tryptic digestion: Beads were resuspended in 400 μl digestion buffer (100 mM Tris, 2 mM CaCl₂, 10% acetonitrile) and 2 μl sequencing grade modified trypsin (Promega) was added and incubated for 16 h at 37 °C. The tryptic peptide fraction was collected, and beads washed twice with 50 mM ammonium bicarbonate to remove all released peptides from beads. The resulting peptide fractions were acidified with 10% formic acid (FA) to pH 3–4 prior to desalting in UltraMicroSpin C18 Columns (Nestgroup) with 5–60 μg capacity for tryptic peptide fraction according to manufacturer's instructions. The eluted peptides were dried in a speedvac and stored at −80 °C.

**LC-MS/MS analysis**. Tryptic peptide fractions and lysates were reconstituted in 20 μL 3% acetonitrile/0.1% FA/doubly distilled water, and 1 ug per sample was loaded onto an EASY-nano-HPLC system (Proxeon) equipped with a RP-HPLC column (75 μm × 10.5 cm) packed in-house with 10 cm stationary phase (Magic C18 AQ 1.9 μm, 200 Å, Michrom BioResources). The HPLC was coupled to a QExactive plus MS (Thermo Scientific) equipped with a nano-electrospray ion source (Thermo Scientific). Peptides were loaded onto the column with buffer A (0.1% FA) and were eluted with 300 nL min⁻¹ buffer B (99.9% ACN, 0.1% FA). Subsequently, the column was washed with 98% buffer B. The tryptic peptide fractions were eluted with a 100 min gradient of 5–20% B followed by a 20 min gradient from 20–28% B, and a final 4-min step from 28–50% B. The column was washed with 98% buffer B. The MS was operated in data-dependent manner, with an automatic switch between MS to MS/MS scans. High-resolution MS scans were acquired (70,000, target value 10⁶) within 300–1700 m/z. The 15 most intense precursor ions were fragmented using higher-energy collisional dissociation (HCD) to acquire MS/MS scans (minimum signal threshold 420, target value 5 × 10⁴, isolation width 1.5 m/z). Unassigned and singly charged ions were excluded from HCD, and dynamic exclusion was set to 30 s.

**Data analysis**. RAW data were converted to mzML using MSconvert. Fragment ion spectra were searched with COMET (v27.0) against UniprotKB (v57.15, Homo sapiens) containing common contaminants. The precursor mass tolerance was set to 20 p.p.m. Carbamidomethylation was set as a fixed modification for cysteine and oxidation of methionine as a variable modification. Probability scoring was done with PeptideProphet and ProteinProphet of the Trans-Proteomic Pipeline (v4.6.2). Protein identifications were filtered for a FDR of ≤1%. For label-free quantification, proteins were filtered for cell surface location, based on the cell surface protein atlas[29] and the human surfaceome (Omasits, U. et al., manuscript in preparation; Supplementary Table 2). The respective ligand was added to the filter list if not contained. For the HATRIC-LRC screen with 1 million cells as starting material, no cell surface filtering was applied. Non-conflicting peptide feature intensities extracted with Progenesis QI (Nonlinear Dynamics). The output of Progenesis is a list of quantified spectral features representing peptides of cell surface proteins with multiple charge states and differential modifications. In MSstats3 (v3.2.2), the features were log-transformed, and then subjected to constant normalization[30]. Protein fold changes and their statistical significance between paired conditions were tested using at least two fully tryptic peptides per protein or one fully tryptic peptide per protein for the 1 million cell experiment. The minimum intensity for each peptide feature was set to 500. Tests for significant changes in protein abundance across conditions are based on a family of linear mixed-effects models. In the last step of the analysis, p values are adjusted for multiple comparisons to control the experiment-wide FDR at a desired level using the Benjamini–Hochberg method. Proteins were considered candidates if they showed a fold-change of 1.5 or higher and an adjusted p value of 0.05 or lower.

**qPCR analysis of siRNA-mediated gene depletion**. qPCR analysis of the efficiency of siRNA-mediated gene knockdown was performed using Power SYBR® Green Cells-to-CT™ Kit (Ambion) and Rotor Gene real-time PCR cycler (Qiagen). The ΔΔCq method was used to determine relative gene expression using HPRT as the housekeeping gene.

**IAV endocytosis assay**. This assay was performed as described in ref.[31]. Briefly, cells were incubated with equal volume of virus (MOI = 50 for the control virus) for 45 min on ice, and the bound virus was allowed to be internalized at 37 °C for 25 min. After washing, the cells were fixed, washed with PBS, and the cell membrane was stained with wheat germ agglutinin (WGA)-AF647 (Invitrogen) in PBS (1:250) for 30 min at RT and washed again to remove unbound WGA-AF647. The epitopes of extracellular HA were blocked overnight at 4 °C with Pinda (anti-HA rabbit polyclonal, made in house) antibody (1:2000) in blocking buffer (PBS, 1%

BSA, 5% FCS), and the cells stained with secondary anti-rabbit IgG-AF594 conjugate (Invitrogen) (1:2500) in blocking buffer for 1 h at room temperature. After fixation in 4% formaldehyde (FA) in PBS for 20 min and washing, a permeabilization solution (PS, 0.1% saponin, 1% BSA, 5% FCS in PBS) was added for 30 min followed by incubation with a mouse monoclonal antibody specific for HA1 (made in house, 1:100) in PS for 2 h at room temperature. After washing, the cells were incubated with secondary anti-mouse IgG-AF488 (Thermo Fisher, A-11001,1:2500) in PS for 1 h, and the nuclei were stained by Hoechst 33258 (Sigma) (1:10,000). Inhibitors against endocytosis mechanisms was used as follows: the cells were pre-incubated with Dyngo-4a (Abcam) (50 μM), which blocks dynamin and/or EIPA (80 μM), an amiloride that blocks Na⁺/H⁺ exchangers, for 30 min prior to infection. HATRIC-coupled or non-coupled virus was submitted to endocytosis assay as described above using equal volumes of both virus samples. The inhibitors were present throughout the duration of the experiment until fixation. Z-stacks were acquired using a Leica SP8 confocal microscope and the internalized virus particles and extracellular particles were quantified using ImageJ.

**IAV infection assay**. This assay was performed as described in ref.[31]. Briefly, cells plated on 96-well μClear cell culture microplates were infected with control or HATRIC-coupled IAV at a low MOI (0.2–0.5) in infection medium (DMEM, 0.2% BSA, 50 mM HEPES pH6.8). 7 h after infection, the cells were washed and fixed in 4% FA in PBS for 20 min at room temperature. The cells were stained using HB65 (anti-nucleoporotein) monoclonal antibody (H16-L10-4R5, ATCC HB-65) from hybridoma supernatant (1:15) in PS for 30 min at room temperature. After secondary antibody and Hoechst staining were performed as above, the plates were imaged using a Yokogawa CV1000 with a 10× objective. The acquired images were analyzed using Cell profiler and an automated pipeline.

**IAV plaque assay**. This assay was performed as described in ref.[31]. Specifically, for IAV titration, HATRIC-coupled and non-coupled virus were infected in Infection PBS (PBS, 0.02 mM Mg²⁺, 0.01 mM Ca²⁺, 0.3% BSA) in 10-fold serial dilutions onto a monolayer of MDCK II cells in 6-well plates. After 1 h of inoculation, the unbound viruses were washed and the medium was replaced with MEM, 0.3% BSA, 0.1% FBS, 20 mM HEPES, 1.2% Avicel, 0.5 μg ml⁻¹ TPCK-trypsin. After 3 days of infection, the viable cells were stained with 0.5 mg ml⁻¹ MTT in PBS for 30 min. Plaques were counted and the plaque-forming unit (PFU) ml⁻¹ was calculated.

**siRNA-mediated depletion of candidates and IAV infection**. A549 cells (2500 cells/well) were reverse transfected with pooled siRNAs (Dharmacon) (4 siRNAs targeting each gene) at a final concentration of 20 nM using Lipofectamine RNAiMax (Invitrogen) in 96-well optical-bottom Matrix plates (Thermo Scientific). Three days after transfection, the cells were infected with IAV strain X-31 (H3N2) at 1.3 × 10⁴ infectious units per well (as determined by TCID₅₀ assay) in infection medium (DMEM, 50 mM HEPES, pH 6.8, 0.2% BSA) for 7 h. IAV X-31 is an H3N2 reassorted strain derived from the A/Puerto Rico/8/34 (PR8) and A/Hong Kong/1/68 strains and was purchased from Virapur in purified form. Infected cells were fixed in 4% formaldehyde in PBS for 20 min. For detection of viral nucleoprotein (NP) expression, cells were permeabilized in 0.1% Triton X-100 for 5 min in blocking buffer (PBS, 1% BSA), stained with anti-NP mAb HB65 (ATCC) for 60 min and then goat anti-mouse Alexa Fluor 488 secondary antibody for 30 min. Nuclei were stained with Hoechst 33258 (Molecular Probes). Automated image acquisition was performed with an ImageXpress high-throughput screening system (Molecular Devices) using a ×10 objective. Cell numbers and raw infection indices for each well were determined using CellProfiler and KNIME software. Infection scores from siRNA-treated samples were normalized to control samples transfected with non-targeting siRNAs. Kruskal–Wallis test was applied to detect significant differences between the groups (p value < 2.2e−16). Statistical significance was tested with a Mann–Whitney U test (Supplementary Data 14) using custom made R-software code.

**Data availability**. All mass spectrometric data that support the findings of this study are deposited to the public database MassIVE as RAW files [ftp://massive.ucsd.edu/MSV000081228]. The authors declare all other data supporting the findings of this study are available within the paper and its supplementary information files.

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

## Acknowledgements

We greatly acknowledge M. Choi and O. Vitek at Northeastern University for help with statistical data analysis. We are grateful to E. Milani, M. Mueller, U. Omasits, S. Mueller and the whole Wollscheid laboratory as well as R. Aebersold for suggestions and support at all stages of the project. This work was supported by funding from Swiss National Science Foundation (SNSF) (#31003A_160259 to B.W.), the InfectX project from the Swiss Initiative in Systems Biology SystemsX.ch (to B.W.) and Commission of technology and innovation (CTI) (to B.W.). Y.Y. gratefully acknowledges SystemsX.ch MRD Project VirX (2014/264) for funding and Dr. Roger Meier from ETHZ for help with statistics. E. M.C. gratefully acknowledges ETH Zurich and the Swiss National Science Foundation (205320_169135) for financial support.

## Author contributions

N.S. and B.W. conceived and designed the study and wrote the primary and subsequent versions of the manuscript; N.S., M.A.S., A.R., A.K., S.G., F. M., Y. Y., E.M.C., and B.W. designed and conducted laboratory experiments; F.M. conducted the experiment with EGF and transferrin on 1 million cells per sample. A.R. and Y.Y. designed and conducted the influenza siRNA treatments and infection assays. M.A.S. and E.M.C. designed and conducted synthesis of HATRIC. All coauthors have reviewed, edited, and approved the manuscript.

## Additional information

**Competing interests:** The HATRIC compound is patented (EP15003213) with N.S., M. A.S., E.M.C., and B.W. listed as inventors. The patent is licensed to Dualsystems Biotech AG (Schlieren, Switzerland) by ETH Zurich. During the time when the experiments were conducted, F.M. was an employee of Dualsystems Biotech AG. The remaining authors declare no competing interests.

