## [Peer Review File · Nature Communications]

Reviewers' comments:

Reviewer #1 (Remarks to the Author):

In this manuscript, the authors describe the synthesis and application of a novel trifunctional crosslinker, HATRIC (clever name, btw), which can be used to identify the cell-surface receptor of orphan ligands. This work builds on their previous work describing a similar crosslinker, TRICEPS. HATRIC has several advantages over TRICEPS including the ability to identify a wider selection of receptors since the HATRIC technology does not rely exclusively on the identification of N-glycopeptides. The authors also describe the use of HATRIC at physiological pH, with addition of a catalyst, compared to their published work with TRICEPS which was performed at pH 6.5. A nice set of applications is shown – including an interesting small molecule application with folate and an application to Influenza A virus.

Receptor identification of orphan ligands remains a challenging area and advancements in this area would be of interest to many bio-researchers. The HATRIC crosslinker itself is quite similar to the TRICEPS reagent previously described – the major functional difference being the replacement of the biotin group for an azide which would allow purification on an affinity resin, without additional protein contamination from streptavidin. HATRIC also has a different protecting group on the hydrazide functional group than TRICEPS, though the authors do not mention whether this has any functional consequences, or was simply a choice made for ease-of-synthesis or other considerations. Several of the major advantages of HATRIC that are highlighted in the manuscript by the authors have been previously described in work on the ASB crosslinker (reference 4 in this manuscript) – the ASB procedure as described also allows identification based on tryptic peptides from the entire protein, rather than focusing on the N-glycopeptides. Although not discussed in detail, the ASB procedure described also appears to use a catalyst and ligand binding is at pH 8.0. Since these appear to be the major advantages cited by the authors for HATRIC, the novelty aspect of HATRIC over TRICEPS may be lessened. There would be definite advantages of HATRIC over ASB – including the simplified ligand labelling and the enrichment using alkyne-beads rather than streptavidin beads. The authors have not described the previous work on ASB in this manuscript, nor compared it to HATRIC.

The manuscript is well-written and clearly presented.

Scientifically and statistically the work presented in this manuscript appears to be generally solid and interesting. However, some details are lacking and the discussion/interpretation of the

experiments and methods is quite limited, perhaps due to space constraints (?).

Specific comments:

1. The description of HATRIC-based ligand-receptor capture in the Materials and Methods section indicates that ligands were incubated with cells at pH 6.5 and does not mention use of a catalyst. This is in direct contradiction to what is discussed in the body of the paper.
2. Are peptides derived from the ligand itself an issue in this method? Would the presence of relatively large amounts of ligand peptides serve to limit loading on the mass spectrometer? If so, this should be mentioned and discussed openly in the manuscript.
3. It appears that the authors have filtered out all proteins that were not on their list of cell surface proteins. Why have they chosen to do this? When this step is not taken, do they find that intracellular proteins are differentially expressed? This filtering step should be mentioned openly in the body of the manuscript and discussed.
4. The implied assertion that HATRIC enables identification of ligands from less cells than TRICEPS is not strongly supported. Both TFR1 and EGFR, that were used in the 1 million cell experiment, are very highly abundant cell surface proteins in MDA-231 cells – is there any data to suggest that the TRICEPS method would not work with 1 million MDA-231 cells with these ligands?
5. If possible, it would be informative to show data for the other catalysts that were tested, so there is more information on why 5-MA was selected. “Evaluation of a number of aniline derivatives led to the identification of 5-MA...”
6. Interpretation of the alternative candidate EGF receptors needs to be handled with some caution. Is there any evidence that some of these ‘candidate receptors’ truly bind to EGF? Is it possible that these proteins may simply be co-localized on the cell surface with the true receptor, leading to enriched proximity-based crosslinking via HATRIC? As written, some biologists may mistakenly take the proteins in Supp Table 1 as ‘proven’ EGF receptors.
7. For the viral work, an interesting follow-on functional study is shown. For this work, the authors should show the level of depletion achieved by the siRNAs for each of these targets. For interpretation of this data, it is important for the reader to know if all of the candidate receptors were successfully depleted and, if so, by how much?
8. Viral mediated entry is a very complicated cellular process, utilizing a wide variety of physiological pathways. I wonder if 21 randomly selected reasonably high abundance cell-surface expressed proteins were chosen for this experiment, would the ‘hit rate’ would be lower than what was seen here? So many proteins would affect one aspect or another of viral entry...

Minor comments:

1. The information provided on the MS results is minimal. While it is great that the MS raw files have been made available, some minimal information should be provided in the manuscript/supplementary info. For example, no peptide-level results are shown or provided. At a minimum, the number of unique peptides identified/quantified for each protein should be

provided in the manuscript. Ideally, some information on the quantitative variability seen between different peptides from the same protein should also be provided.

2. More details on the statistical methods used would be helpful. How were protein-level p-values determined? How was quantitative data from individual peptides combined? How were the different technical replicates used for this calculation? What modules from MSstats were used?

3. Methods section: pH required for digestion buffer description.

4. The main body text describing the 1 million cell experiment should mention that this was in MDA-231 cells.

Reviewer #2 (Remarks to the Author):

The manuscript by Sobotzki et al. describes a novel technique to fish for cellular receptors for a variety of ligands, including proteins, small molecules and viruses. The authors developed a trifunctional organic compound, which can covalently link proteinaceous ligands and through a second reactive group covalently link receptor molecules after incubation of the compound-ligand molecule with target cells. Finally a third reactive group allows the purification of putative receptor-ligand-compound complexes by click chemistry. Purified proteins are quantified by LC-MS/MS analysis using standard protocols and compared to control conditions, in which receptor binding of the compound labeled ligand is competed for with an excess of unlabeled ligand or the compound is rendered inactive by glycine quenching. The authors perform four proof-of-principle experiments. First they use the method to confirm epidermal growth factor (EGF) binding to EGF receptor (EGFR). Second, they determine the experimental threshold using transferrin binding to its cognate receptor. Third, the authors demonstrate applicability to the small organic ligand folate. Lastly, they perform an experiment with influenza A virus bound to human lung epithelial cells. While the compound synthesis and the first three proof-of-principle experiments are well designed and controlled, the experiments on influenza virus require some attention. Moreover I recommend a thorough discussion of the discrepancy between the identified IAV attachment factors and previously described host factors (multiple RNAi screens). Also the false positive rate should be discussed as detailed below. Overall, the manuscript is however very well written and the description of methods is clear to non-expert readers. Statistical analysis of MS data is sound. Once the points below are addressed, I favor publication of this description of an exciting and promising new technology, which is clearly of interest to various fields of biology.

Major comments:

1. Fig. 2e. Why was insulin used as control ligand? While the first three experiments were well

controlled, this control seems random. New datasets with compound-free virus competition or quenched virus would seem better controls.

2. Fig. 2e. Compound labeling of viruses can strongly affect infectivity. The authors should perform control experiments, in which they compare titers of virus before and after labeling. Moreover the effect of labeling on the specific infectivity (infectious particle / genome copy number) should be measured.

3. Fig. 2e. Compound labeling of small enveloped viruses such as influenza A virus may affect its entry route. The authors should experimentally demonstrate that the entry pathway into A549 cells is not altered after compound labeling of the virus particles using inhibitory compounds and/or imaging techniques.

4. Fig. 2 and lines 316-331: The authors should explain the filtering for cell surface molecules in the main manuscript, not only in the methods. They should disclaim, which fraction of the identified proteins was cell surface associated according to e.g. GO annotation.

5. Fig. 2e. Why was a nuclear pore protein (NUP210) identified despite the surfaceome filtering? What is the leakiness of the method towards cytoplasmic or nuclear proteins?

6. Line 177: Multiple RNA interference (RNAi) screens on influenza have been published, with some overlap. It is recommended that the authors discuss in more detail why on the one hand the published RNAi hits were not discovered in their HATRIC experiment and on the other hand, why their MS hits were vice versa not previously identified in any of the influenza host factor searches.

7. A differentiated discussion on the limitations of the technology is missing. Can any small ligand be linked to the HATRIC compound without affecting receptor affinity? What are the requirements of organic compounds to be successfully fused to HATRIC by synthesis?

8. The full MS datasets should be disclosed in supplementary tables and deposited in public online repositories such as the EMBL/EBI IntAct database. In particular for the influenza A virus experiment.

Minor comments:

1. Full protein names are not mentioned. Please write out the full names at first mentioning of a protein abbreviation, such as FOLR1.

2. Supplementary table 3: The human surfaceome should be presented with separate columns for gene name, protein name and Uniprot accession number for easier accessibility.

3. Fig. 2e,f. The gene/protein names do not match between Fig. 2e, Fig. 2f, Tab. S2 and Tab S4. If the authors decide to use protein names in Fig. 2e and gene names in Fig. 2f, it is advisable to include both – protein names and gene names – in Tab S2 and S4 to allow the reader to match the datasets.

4. Certain proteins, which were silenced (Fig. 2f), are not included in Tab. S2 or annotated differently. Examples are SLC19A1, NUP210, ABCC4.

Reviewer #3 (Remarks to the Author):

In the manuscript from Sobotzki et al., the authors demonstrate their development of next-generation LRC method. Having been the leading developers of the first-generation reagents, TRICEPS-LRC, the Wollscheid laboratory is well-suited to evolve this useful technology for improved coverage, applicability, and sensitivity. The updated methodology, termed HATRIC, still employs the key step of receptor sugar alcohol to aldehyde periodate oxidation, and subsequent coupling to the hydrazine-containing probe. However, the authors optimized the periodate oxidation to achieve high efficiency at neutral pH. In addition, the authors introduced Click chemistry in the HATRIC reagent. These optimizations directly contribute to the improved sensitivity of the approach, with a minimum requirement of between 1 -2 orders of magnitude less cellular material. The authors experimentally demonstrated the results of HATRIC-LRC with 1 million cells, though as mentioned in the comments below, the explanation of this experiment in the manuscript could be improved. The work nicely demonstrates the broad application of the method to a range of ligands, including the small molecule folate, the polypeptide EGF, and the intact virus, influenza A. The authors convincingly demonstrated that their technology could identify biologically relevant cell surface receptors of IAV by validation with siRNA knockdown of candidate IAV cell surface receptors during infection. However, as mentioned in the main comments section, the authors did not fully discuss why none of the known IAV receptors were identified.

Overall, this is a strong methodological study with significant application to biomedical and pharmaceutical research, particularly in contributing to the characterization of orphan receptors. The authors do have a few outstanding and several minor points to address; however, if these can be addressed, I would recommend the manuscript for publication.

Main Points

1. A general main point is the lack of discussion related to novel identified candidates or lack of identification for known candidates in the case of IAV. For instance, in addition to identifying the known receptors for the EGF and folate ligands, the authors found several other putative candidates, which the authors did not discuss. What percent were known or predicted cell surface or secreted proteins? In addition, for the IAV experiments, the authors state: " We identified 24 virus-interacting candidates (Fig. 2e, Supplementary Table 2)." Before discussing the siRNA results, the authors should expand on their statement. Later in the manuscript, the authors mention that none have been previously implicated. However, it might be appropriate for the authors to briefly discuss here, (1) that these targets didn't include the known receptors, (2) how many known receptor targets are there for IAV, (3) their thoughts on why HATRIC did not capture them?

2. Did the authors evaluate intracellular generation of aldehydes with the improved periodate oxidation using 5-MA? Is the HATRIC reagent cell permeable, e.g. with a small molecule conjugate like folate?

3. The overall strategy and figure panel (Fig 2b) to identify “EGFR as the receptor for anti-EGFR antibody and transferrin receptor protein 1 (TFR1) as the receptor for Holo-transferrin (TRFE) from 1 million cells per sample” is confusing. The idea of testing the limit of detection for HATRIC (1 million cells) is clear, but how is this related to anti-EGFR antibody? Is this used instead of HATRIC? What is the relationship between EGFR and TRFE? This experiment should be described in the Methods section.

Minor Points

1. The first description of HATRIC in Fig 1b, has an application that is targeted to specific glycoproteins or glycoprotein classes using ligand coupling. Although the first generation of TRICEPS was also a LRC method, could HATRIC (and in general these technologies) be used to gain broad capture of the glycoproteome in the absence of ligand coupling.

2. In general for LRC technologies, is ligand-receptor activation and receptor-mediated events such as internalization an issue?

3. The authors state: “The novel workflow renders HATRIC-LRC independent of the PNGase F deglycosylation reaction, ultimately enabling a more robust relative quantification of cell surface receptors than is possible with first-generation LRC”. This seems to imply that the first-generation LRC (assume TRICEPS-LRC) could not be performed without PNGaseF. If TRICEPS-peptide capture was performed (as in the authors previous work), then I would agree. However, couldn't TRICEPS-LRC be performed with a protein capture, as described for HATRIC, which would allow bead-based digestion as well?

4. Conceptual flow of Figure 1b needs improvement. In the text, the description of steps follows from (1) periodate oxidation to (2) addition of HATRIC-LRC, but in Fig 1b, the periodate step is not explicit until the second box, which is after HATRIC-LRC/arrow graphic. The authors should illustrate the periodate oxidation step and resulting modifications explicitly, before addition of HATRIC-LRC?

5. The authors could consider integration the chemical structure of the catalyst 5-methoxyanthranilic acid (Fig 1c) into Fig 1d, perhaps as a mini-graphic next to the dashed trace, or alternatively, into the supplement.

6. In volcano plots for Fig 2, since there are a limited number of significant candidates, the authors should consider labeling all points with gene symbols/arrows, as needed.

7. For the IAV experiment, what was the rationale for choosing insulin as a control instead of quenched HATRIC? I assume this was a positive control? If so, this should be explained more explicitly. Given the authors employ several options for controls, a few sentences clarifying the practical selection of controls could be helpful, especially regarding the above two options. For

instance, if the positive control and experimental condition share a receptor, then the ratio would be 1:1 and eliminated from consideration.

8. In Figure 2f, what is an infection score? If it has units, it should be defined in the legend.

9. Include units of concentration on the x-axis in Supplementary Fig 1.

10. In the Tables, the authors should check their gene names for accuracy. For instance, in Table S1, the entries P09110 and O15427, the genes listed do not match the UniProt annotated genes.

Point-by-point response

Reviewer #1 (Remarks to the Author):

- In this manuscript, the authors describe the synthesis and application of a novel trifunctional crosslinker, HATRIC (clever name, btw), which can be used to identify the cell-surface receptor of orphan ligands. This work builds on their previous work describing a similar crosslinker, TRICEPS. HATRIC has several advantages over TRICEPS including the ability to identify a wider selection of receptors since the HATRIC technology does not rely exclusively on the identification of N-glycopeptides. The authors also describe the use of HATRIC at physiological pH, with addition of a catalyst, compared to their published work with TRICEPS which was performed at pH 6.5. A nice set of applications is shown – including an interesting small molecule application with folate and an application to Influenza A virus.

- We would like to thank this reviewer for the very good summary emphasizing the advantages of the HATRIC-based LRC compared to the TRICEPS-based LRC technology which enabled the discovery of receptors involved in Influenza infection.

- Receptor identification of orphan ligands remains a challenging area and advancements in this area would be of interest to many bio-researchers. The HATRIC crosslinker itself is quite similar to the TRICEPS reagent previously described – the major functional difference being the replacement of the biotin group for an azide which would allow purification on an affinity resin, without additional protein contamination from streptavidin. HATRIC also has a different protecting group on the hydrazide functional group than TRICEPS, though the authors do not mention whether this has any functional consequences, or was simply a choice made for ease-of-synthesis or other considerations. Several of the major advantages of HATRIC that are highlighted in the manuscript by the authors have been previously described in work on the ASB crosslinker (reference 4 in this manuscript) – the ASB procedure as described also allows identification based on tryptic peptides from the entire protein, rather than focusing on the N-glycopeptides. Although not discussed in detail, the ASB procedure described also appears to use a catalyst and ligand binding is at pH 8.0. Since these appear to be the major advantages cited by the authors for HATRIC, the novelty aspect of HATRIC over TRICEPS may be lessened. There would be definite advantages of HATRIC over ASB – including the simplified ligand labelling and the enrichment using alkyne-beads rather than streptavidin beads. The authors have not described the previous work on ASB in this manuscript, nor compared it to HATRIC.

- We would like to thank the reviewer for the valuable suggestion to add information about similarities and differences compared to ASB. In principle, it is very good for the community that complementary technologies are

46 available to decode ligand receptor interactions. There is a wealth of ligands
out there in search for receptors and having different strategies and
chemistries available is certainly of advantage for the community. The
HATRIC-based LRC strategy is indeed a protein-based workflow and this is
similar in parts to the ASB strategy. However, the chemistry used for the
HATRIC-based approach is novel and makes the difference. The next
generation HATRIC sporting the acetone-protected hydrazide functionality in
combination with click-chemistry and the catalyst, allowing for reactions in
different ligand receptor interaction suitable pH ranges, enables now new
applications and delivers results with unprecedented sensitivity, as shown in
the manuscript. Furthermore, this new combination of chemistries within the
HATRIC-LRC workflow allows for the first time a significant reduction of
cellular starting material needed for the discovery of receptors compared to
ASB and TRICEPS-based LRC workflows. HATRIC-LRC can be routinely
performed with 1x 150mm dish vs. 5-7x 150mm plates in ASB and 4x 150mm
plates in TRICEPS-LRC. In addition, the catalyst-enhanced HATRIC-LRC
never required us to increase the sodium periodate concentration beyond 1.5
mM (compared to up to 10mM in ASB) which is a clear advantage in respect
to cell viability during the process of labeling, especially with primary cells.
Finally, HATRIC-LRC - for the first time - enabled the receptor
capture/identification with a small molecule compound which was never
before demonstrated on cell surface proteins.

○ We added new text as detailed below to the introduction and
discussion section and after completion of the suggested edits, the
revised manuscript has benefitted from an improvement in the overall
presentation and clarity.

- ● Regarding your comment related to the functional consequences of changing
the hydrazide protection group in HATRIC we would like to provide you with
more context and insights. Investigating the pH as a critical factor during the
receptor capture reaction, we tested the impact of different protection groups
on the yield of hydrazone formation on live cells at higher pH (pH 7.6). We
employed the first generation of TRICEPS compounds bearing a NHS group
coupled to a biotin and a hydrazide group and studied two different TRICEPS
versions bearing either a trifluoroacetyl-protected (**PbP Figure 1A**) or
acetone-protected (**PbP Figure 1B**) hydrazide. When comparing hydrazone
formation of these two TRICEPS versions on the cell surface, we detected

much brighter cell surface labeling with the acetone-protected hydrazone-
 containing compound compared to the Tfa-protected under the same
 conditions (visualized by Streptavidin-FITC) at both pH 6.5 and pH 7.6 on live
 A2.01 cells (**PbP Figure 1C**). These experiments, conducted in the absence
 of the catalyst, indicate higher reactivity in the cell surface micro-environment.
 The possibility to conduct the experiments at different pH levels, supported in
 addition kinetically by the the catalyst, turned out to be a major advantage for
 studying pH-sensitive ligand-receptor interactions, such as between folate
 and folate-receptor alpha: Folate-based receptor capturing was never
 successful at pH 6.5, but only at pH 7.4.

**PbP Figure 1** | Flow cytometric comparison of pH-dependent hydrazone formation of
 glycine-quenched TRICEPS bearing two different protection groups: the original Tfa-
 protection group (**A**) or the new acetone-based protection group (**B**) on A2.01 cell line. Cells
 were oxidized with 1.5mM NaIO₄.

- • **Changes to the manuscript:** “Ligand-based receptor capture (LRC)
 technology partly overcame these difficulties and enabled the identification of
 ligands for orphan N-glycoprotein-receptors using the tri-functional reagent
 TRICEPS (Frei et al. 2012, 2013) and modifications thereof in ASB (Tremblay
 and Hill 2017). Application of TRICEPS-LRC and ASB in different biological
 systems, however, revealed the need to redesign the first-generation

technologies: TRICEPS-LRC was intentionally designed to enable the
identification of ligand-bound receptors solely based on formerly N-
glycosylated peptides. O-glycosylated receptors and N-glycosylated receptors
whose deamidated peptides were not detectable by mass spectrometry were
eventually missed by this strategy. However, this peptide-based strategy
benefitted from the ability and quality to be able to filter for deamidated
receptor peptides as indicators of direct TRICEPS-crosslinking and ligand-
binding. In contrast, in ASB, tryptic digestion is performed directly on
Streptavidin beads, which enables protein-level affinity purification, enabling,
in principle, the identification of receptors through non-glycopeptides.
However, direct digestion of proteins bound to Streptavidin beads leads to
major contaminations with streptavidin peptides, impairing identification and
label-free quantification of receptor peptides. Furthermore, ASB requires
performing a two-step reaction in order to couple the ligand to the cross-
linker, and biotin transfer from ligand to receptor is mediated by reduction of a
disulfide bond, making its application sensitive to reductive environments.
Furthermore, the ASB strategy utilizes a catalyst to catalyze oxime formation
on the cell surface at pH 8. Similar to first generation TRICEPS-LRC, ASB
requires high amounts of starting material (50 million cells or 5-7 150mm
plates) and captures ligand-receptor interactions at pH 8 compared to pH 6.5
for TRICEPS LRC. The pH of the microenvironment directly influences the
affinity between a ligand and its receptor, exemplified by ligands that are
internalized upon receptor binding: The affinity for the receptor is high at pH
7.4 on the surface of living cells, but decreases upon acidification (pH 6.5) in
the endosome, leading to release of the ligand from the receptor. A prime
example of this is folate, which has an affinity for folate receptor alpha
(FOLR1) that is 2000 times lower at pH 6.5 than at pH 7.4 (Yang et al. 2007).
Consequently, the folate receptor has not been detected by TRICEPS-LRC in
the past, highlighting the need for a next-generation LRC suited for receptor
deorphanization at physiological pH. [...]

To enable HATRIC-LRC under physiological conditions, it was necessary to
accelerate the reaction of hydrazines with aldehydes, which is slow at neutral
pH (Dirksen and Dawson 2008). Aniline has been exploited to catalyze similar
reactions efficiently (Bhat et al. 2010), however, the cytotoxicity at the
required concentration limits use with live cells (Khan et al. 1999). Aniline-
derived water-soluble catalysts have been described that substantially
improve catalysis of hydrazone formation, but none had been tested in

biological systems (Crisalli and Kool 2013). Evaluation of a number of aniline
derivatives regarding their solubility, cytotoxicity and capability to enhance
hydrazone formation between aldehydes on cell surface proteins and the
HATRIC-hydrazide on living cells led to identification of 5-methoxyanthranilic
acid (5-MA, **Fig. 1c**, **Supplementary Fig. 1**). 5-MA catalyzed hydrazone
formation at a non-toxic concentration at pH 7.4 more efficiently than 2-
amino-4,5-dimethoxy benzoic acid (ADA). Additionally, replacing the original
Trifluoroacetyl-protection group of TRICEPS by an acetone-derived protection
group in HATRIC enabled higher yield of hydrazone formation on live cells
(**data not shown**). Last, we confirmed that under the chosen conditions,
HATRIC does not penetrate cells avoiding contamination with intracellular
proteins (**Supplementary Fig. 2**).

The manuscript is well-written and clearly presented.

- • Thank you very much & the comment is very well appreciated.

Scientifically and statistically the work presented in this manuscript appears to be generally
solid and interesting. However, some details are lacking and the discussion/interpretation of
the experiments and methods is quite limited, perhaps due to space constraints (?).

- • The lack of some details is mainly due to the initial space constraints of the
format. We now added more details in the text and in the supplementary
information.

Specific comments:

1. The description of HATRIC-based ligand-receptor capture in the Materials and
Methods section indicates that ligands were incubated with cells at pH 6.5 and does not
mention use of a catalyst. This is in direct contradiction to what is discussed in the body of
the paper.

- • Thank you for noticing, this was indeed rectified as suggested by the
reviewer.
- • **Changes to the methods section:** Cells were washed once with PBS (pH
6.5) and resuspended in 10ml PBS containing 5mM 5-MA (pH 7.4).

2. Are peptides derived from the ligand itself an issue in this method? Would the
presence of relatively large amounts of ligand peptides serve to limit loading on the mass
spectrometer? If so, this should be mentioned and discussed openly in the manuscript.

- • In theory, large amounts of ligands bound to the cell surface via the HATRIC
compound could potentially cause problems, depending on the speed and

dynamic range of the MS instruments used for analysis. However, peptides
from the ligand itself never presented an issue in our hands. Sample
complexity remained low in HATRIC-based experiments and the improved
speed and sensitivity of the latest Orbitrap instruments (QE, FUSION &
LUMOS) enabled straightforward sample analysis.

3. It appears that the authors have filtered out all proteins that were not on their list of
cell surface proteins. Why have they chosen to do this? When this step is not taken, do they
find that intracellular proteins are differentially expressed? This filtering step should be
mentioned openly in the body of the manuscript and discussed.

• This is correct and can be explained. The primary output of a HATRIC-LRC
screen is a list of quantified spectral features representing all proteins
identified in such an experiment. In this list, a number of proteins is identified
that are not annotated to reside at the cell surface and/or do not contain
transmembrane domains (we refer to this fraction as “nonspecific” proteins).
We investigated several sources of this background but couldn’t determine
the source and can thus only speculate about technical reasons why these
proteins are identified in HATRIC-LRC screens, similar to other screening
technologies. For the purpose of selecting receptor candidates for further
validation, one can, in principle, directly quantitatively compare protein
abundances of all identified proteins without any filtering. The quantitative
comparison will help to hide the majority of “unspecific” proteins in the scatter
plot as not specifically enriched, as these are somewhat equally identified
across samples. This approach may be sufficient to identify highly abundant
or large cell surface proteins or cell surface proteins that are highly
soluble/MS detectable, but it highly neglects proteins that are small, of lower
abundance or have many, hardly soluble transmembrane-spanning peptides.
It is well known that cell surface proteins are notoriously difficult to identify by
MS and our strategy enables the identification of hundreds of cell surface
proteins using a chemoproteomic strategy. Therefore, this approach is
inadequate when one is interested in these typically underrepresented
species. To increase the informative value of such screens, we recommend to
filter HATRIC-LRC data sets with our surfaceome filter to enable the
identification of low abundant proteins that are typically overlooked and push
them over the significance value against the background of “nonspecific”
proteins with many peptides. **Taken together, filtering doesn’t change the**
**fold changes of proteins across samples, but significantly affects p-**

**values.** At the same time, the screening protocol is by no means 100%
efficient and considerable losses of peptides are expected during glycan
oxidation, aldehyde capturing, affinity purification and tryptic peptide release,
as well as peptide purification. Therefore, in our experience, the “nonspecific”
peptides are essential to “chaperone” the membrane protein-derived peptides
to the MS. Further, we would like to point out politely that filtering is commonly
performed in screens, such as filtering for proteins that are identified with a
minimum number of peptides (ASB) or that carry specific sequence motifs
such as the N[115]-X-S/T signature in TRICEPS-LRC. In cases, where no cell
surface filter list is available (e. g. more exotic mammals), we recommend to
include one further step in the protocol and release N-glycosylated peptides
from the beads using PNGase F and limit quantification to proteins that were
identified in the N-glycopeptide fraction.

● All of this is best exemplified in **Pbp figures 2 and 3** where the virus and EGF
data were left unfiltered prior to statistical analysis in MSstats 3.2.2. In the
virus data analysis, two of our most promising receptor candidates, namely
PLD3 and APMAP remain below significance level and would not have been
further investigated (**PbP Figure 2**). However, in our follow-up experiments,
both proteins showed promising evidence to impact viral entry. In the
unfiltered experiment 2132 proteins were quantified in the virus and insulin
sample, whereas our cell surface filtering left 213 proteins for quantitative
analysis. Similar effects were observed for EGF (**PbP Figure 3**) where EGF
remained below the significance cut-off even though we know that it was
more abundant in the EGF sample. Interestingly, for the HATRIC LRC with 1
million cells as starting material, no further filtering was required as the lower
amount of cellular starting material lead to higher specificity in the sample,
where 34% of proteins were already annotated as cell surface proteins
(according to our surfaceome filter list).

PbP Figure 2 | Volcano plot from H3N2-based HATRIC-LRC on 20 million A549 cells without applying the surfaceome filter list prior to quantitative data analysis.

PbP Figure 3 / Supplementary Fig. 3 | Volcano plot from EGF-based HATRIC-LRC on 20 million H358 cells without applying the surfaceome filter list prior to quantitative data analysis.

- **Changes to the manuscript:** [...] Trypsin-mediated proteolysis of bead-bound proteins releases the un-glycosylated peptides. These peptides are analyzed with high-accuracy mass spectrometry using data-dependent acquisition and filtered for known and predicted cell surface proteins. The quantitative comparison to the competitive control reaction reveals specific enrichment of target cell surface receptors for the ligand. [...] We validated HATRIC-LRC demonstrating capture of epidermal growth factor receptor (EGFR) using epidermal growth factor (EGF) as a ligand in an experiment with live H-358 cells (Fig. 2a). When quantifying all identified proteins across samples, we found 9 proteins significantly enriched in the EGF-captured samples, but only three of them were cell surface proteins, and EGF as ligand dropped below significance level. Statistical scoring of protein candidates is based on the number of peptides identified per proteins which leads to bias towards larger proteins or proteins whose peptides are easily detectable in MS (e. g. 19 features were quantified and scored statistically for EGFR, whereas only 1 peptide was quantified and scored for EGF). In order to overcome this bias, we used a filter for known and predicted cell surface proteins prior to statistical scoring to rescue receptor candidates where most peptides are hardly detectable via MS (e. g. due to decreased solubility) (**Supplementary Fig. 3, Supplementary Table 1**). Applying this filter prior to statistical analysis, we correctly identified EGF significantly enriched and identified five other EGF receptor candidates that have not been described before (**Supplementary Table 3**), namely monocarboxylate transporter 4 (SLC16A3), filamin-A (FLNA), peroxisomal 3-ketoacyl-CoA thiolase (ACAA1), transmembrane emp24 domain-containing protein 7 (TMED7) and sarcoplasmic/endoplasmic reticulum calcium ATPase 1 (AT2A1) (**Supplementary Table 3**). Reports of direct interactions between these proteins and EGF are not available, but it was shown before that SLC16A3 co-locates with CD147 in breast cancer cells (Gallagher et al. 2007), which in turn is associated with EGFR in similar lipid domains (Vial and McKeown-Longo 2012) suggesting that SLC16A3 resides in the neighbourhood of EGFR at the cell surface (Dai et al. 2013). [...]
- **Changes to the methods section:** For label-free quantification, proteins were filtered for cell surface location, based on the cell surface protein atlas (Bausch-Fluck D. et al. 2015, PLoS One 10: e0121314) and the human

surfaceome (Omasits U. et al., manuscript in preparation; **Supplementary**
**Table 2**). The respective ligand was added to the filter list if not contained in
the database. For the HATRIC-LRC screen with 1 million cells as starting
material, no cell surface filtering was applied. Non-conflicting peptide feature
intensities extracted with Progenesis QI (Nonlinear Dynamics).

4. The implied assertion that HATRIC enables identification of ligands from less cells
than TRICEPS is not strongly supported. Both TFR1 and EGFR, that were used in the 1
million cell experiment, are very highly abundant cell surface proteins in MDA-231 cells – is
there any data to suggest that the TRICEPS method would not work with 1 million MDA-231
cells with these ligands?

● We tried to identify EGFR and TFR1 using anti-EGFR antibody and holo-
transferrin (hTF) on 1 million MDA-MB-231 by TRICEPS-LRC, but failed
repeatedly (**PbP Fig. 5, PbP Fig. 7**). In parallel, we conducted TRICEPS-LRC
on 50 million MDA-MB-231 cells and successfully identified EGFR as receptor
for EGF (**PbP Fig. 4, PbP Fig. 6**). However, we were also not able to identify
TFR1 for receptor of hTF in this particular experiment. This might be
explained by the fact that transferrin is released from the cell at pH 5.5
making the experimental setup with transferrin prone to failure in the low pH
setting of TRICEPS-LRC. When conducting the same experimental setup
using insulin and EGF as ligands, we were only able to identify the
corresponding receptors on 50 million cells and identified none of the
receptors with 1 million cells as starting material. In all experiments, we used
the originally published experimental conditions to perform TRICEPS-LRC
(Frei et al. 2013).

● These experiments highlight the difficulty to identify receptors solely based on
N-glycopeptides with the original TRICEPS LRC in a reliable and reproducible
fashion from lower amounts of cells, even if the receptors are of high
abundance on this particular cell line. These experiments just serve as
examples for a larger number of experiments that we conducted in our
laboratory pointing in the same direction. Due to the new chemistry and
workflow used in HATRIC-based LRC workflows we do now have the
opportunity to deorphanize ligands and detect their receptor(s) from as little
as one million cells.

● We added PbP Figures 4&5 to the supplement (**Supplementary Fig. 5**).

PbP Figure 4 / Supplementary Figure 5A | Volcano plot from anti-EGFR antibody- and holo-transferrin-based TRICEPS-LRC on 50 million MDA-MB231 cells.

PbP Figure 5 / Supplementary Figure 5B | Volcano plot from anti-EGFR antibody- and holo-transferrin-based TRICEPS-LRC on 1 million MDA-MB231 cells.

PbP Figure 6 | Volcano plot from anti-EGFR antibody- and insulin-based TRICEPS-LRC on 50 million MDA-MB231 cells.

PbP Figure 7 | Volcano plot from anti-EGFR antibody- and insulin-based TRICEPS-LRC on 1 million MDA-MB231 cells.

- **Changes to the manuscript:** As HATRIC-LRC is based on protein-level purification, more than one peptide is commonly identified per protein, such as exemplified by EGFR (**Supplementary Fig. 4**). Therefore, we investigated the HATRIC-LRC detection limit with respect to the amount of starting material needed for successful receptor identification. From as little as one million MDA-MB-231 cells per sample, we were able to unambiguously

identify EGFR as the receptor for HATRIC-coupled anti-EGFR antibody and
transferrin receptor protein 1 (TFR1) as the receptor for HATRIC-coupled
Holo-transferrin (TRFE) (**Fig. 2b**) which was not possible with TRICEPS-LRC
(**Supplementary Figure 5, Supplementary Table 6**). Where possible, we
recommend the usage of of 5-20 million cells in order to detect low copy
number receptors based on a given sensitivity of the MS instrument used for
analysis.

5. If possible, it would be informative to show data for the other catalysts that were
tested, so there is more information on why 5-MA was selected. "Evaluation of a number of
aniline derivatives led to the identification of 5-MA..."

• We identified four potentially relevant catalysts in the literature and in
discussions with the Carreira group at ETH: aniline, 2-amino-4,5-dimethoxy
benzoic acid (ADA), 3-amino-2-naphthoic acid (ANA) and 5-
methoxyanthranilic acid (5-MA). We excluded p-phenylenediamine very early
due to suggested oxidative instability and toxicity (Kool, Chem Rev, 20017,
117, 10358 as well as Kool, ACS Chem Biol 2016, 11, 2312). First, we
investigated water solubility in PBS: All tested compounds were soluble in
PBS at least up to a concentration of 100mM with exception of ANA (fully
soluble up to 1mM only with 0.2% DMSO) and was therefore excluded from
further analysis. We executed alamarBlue™ cytotoxicity assays to determine
cell viability at catalytically relevant concentrations (**PbP Figure 8**). Avoiding
cytotoxicity is essential to HATRIC-LRC, as disrupting cellular integrity would
lead to unwanted labeling of intracellular proteins. Upon cytotoxicity testing,
we excluded aniline for the highest cytotoxicity. As 5-MA is a derivative of
anthranilic acid, a substrate in the tryptophan biosynthesis, cytotoxicity was
expected to be reduced compared to aniline. However, these findings were
never confirmed experimentally for reactions on live cells. This is the first time
reported that 5-MA was used on live cells where no cytotoxic side effects
were observed and hydrazone formation was catalyzed.

 **PbP Figure 8 / Supplementary Figure 1** | Cytotoxicity of aniline and aniline-derived
 organocatalysts on MDA-MB 231. MDA-MB 231 cells (20.000 cells/well in a 96-well plate)
 were treated with the indicated concentrations of catalyst in DMEM (pH adjusted to 7.4, 1%
 Pen/Strep) for 1.5h at 37°C. Supernatant was replaced by 100ul DMEM with 10%
 alamarBlue™ reagent (ThermoScientific) and incubated for 5h at 37°C in the dark. Assay
 was read out by a fluoreader (Ex: 545nm, Em: 590nm, automatic gain).

- We tested both ADA and 5-MA in the flow cytometric experiment presented

 (Fig. 1C) for catalysis of hydrazone formation on live cells. 5-MA showed the
 highest catalytic effect in a HATRIC-LRC, as assessed by FACS. The
 difference between 5MA and ADA was small, but reproducible and led to the
 decision to use 5-MA in all future experiments.

 **PbP Figure 9 / Figure 1C** | Flow cytometry traces of U-2932 cells incubated with HATRIC
 conjugated to dibenzocyclooctyne-Alexa Fluor 488 (DIBO-AF488) at pH 6.5 or pH 7.4 in the
 presence or absence of organocatalyst 5-methoxyanthranilic acid (5-MA) (Structure shown,
 Mw = 167.16 g/mol) or 2-amino-4,5-dimethoxy benzoic acid (ADA). HATRIC was quenched

with glycine (Gly-) to avoid potential reaction of HATRIC's NHS-ester with aminogroups at
the cell surface. Shift to the right indicates more efficient labeling with HATRIC-DIBO-AF488.

• **Changes to the manuscript:** We included both figures (cytotoxicity and
FACS) and changed the figure legend as follows: [...] Evaluation of a number
of aniline derivatives regarding their solubility, cytotoxicity and capability to
enhance hydrazone formation between aldehydes on cell surface proteins
and the HATRIC-hydrazide on living cells led to identification of 5-
methoxyanthranilic acid (5-MA, **Fig. 1c, Supplementary Fig. 1**). 5-MA
catalyzed hydrazone formation at a non-toxic concentration at pH 7.4 more
efficiently than 2-amino-4,5-dimethoxy benzoic acid (ADA). [...]

• **Figure Legend 1C:** [...] Flow cytometry traces of U-2932 cells incubated with
HATRIC conjugated to dibenzocyclooctyne-Alexa Fluor 488 (DIBO-AF488) at
pH 6.5 or pH 7.4 in the presence or absence of organocatalyst 5-
methoxyanthranilic acid (5-MA) (Structure shown, Mw = 167.16 g/mol) or 2-
amino-4,5-dimethoxy benzoic acid (ADA). [...]

• **Changes to the materials and methods section:**

○ **Catalyst Cytotoxicity Assays:** MDA-MB 231 cells (20.000 cells/well
in a 96-well plate) were treated with the indicated concentrations of
catalyst in DMEM (pH adjusted to 7.4, 1% Pen/Strep) for 1.5h at 37°C.
Supernatant was replaced by 100ul DMEM with 10% alamarBlue™
reagent (ThermoScientific) and incubated for 5h at 37°C in the dark.
Assay was read out by a fluoreader (Ex: 545nm, Em: 590nm,
automatic gain).

○ **FACS:** [...] Cells were labeled with 75 µM glycine-quenched HATRIC-
DIBO-AF488 conjugates for 60 min at 4 °C with slow rotation in the
presence or absence of 5 mM 5-MA or 5mM ADA.

6. Interpretation of the alternative candidate EGF receptors needs to be handled with
some caution. Is there any evidence that some of these 'candidate receptors' truly bind to
EGF? Is it possible that these proteins may simply be co-localized on the cell surface with
the true receptor, leading to enriched proximity-based crosslinking via HATRIC? As written,
some biologists may mistakenly take the proteins in Supp Table 1 as 'proven' EGF
receptors.

• HATRIC-LRC is a screening technology which enables the identification of
receptor candidates. In certain case scenarios, identified candidates may not
be direct interaction partners of the ligand as you pointed out. Apart from the

main receptor, other candidates identified could be “next door neighbours”,
potentially influencing receptor activity, which were captured due to proximity
to the main receptor. We are following up on this exciting possibility. Given
the experimental setup, the candidates identified from HATRIC-LRC
experiments can generally be the result of four reasons: (1) there is a direct
interaction of the ligand with the target receptor; (2) the protein is in close
proximity of the target receptor (“neighbourhood protein”); (3) the protein gets
upregulated in response to treatment with the ligand and gets
overrepresented in the background binding of HATRIC (e. g. we use
approximately 8 times more EGF than is used for stimulation experiments) or
(4) the identified candidate is a false positive. Our experiments do not allow
460 us to delineate right away which type of interaction was observed, but the
461 validation experiments and the cited data clearly underline the relevance of
462 the identified proteins. The analysis pipeline was optimized to allow for
identification and ranking of receptor candidates. However, the resulting data
have to be analyzed carefully and more stringent receptor spaces can be
defined based on the identification of positive control receptors or the ligand
(e.g. EGF). Identified candidates need validation in tailor-made follow-up
experiments, such as siRNA-based approaches. These approaches cannot
be generalized and for every LRC application the type of follow-up experiment
will depend on the type of ligand, the biological context, and the tools
available for the system under study. However, we would also like to point out
that the biological relevance of the neighbouring proteins is not to be
underestimated either. Proteins that are in close proximity of the target
receptor might interfere with the activity of the actual target and are therefore
relevant for future studies of the lateral cell surface interactome. HATRIC-
LRC could potentially also be used to generate candidates for such studies -
another exciting application of HATRIC-LRC for life science research.

- • **Changes to the manuscript:** [...] Applying this filter prior to statistical
analysis, we correctly identified EGF significantly enriched and identified five
other EGF receptor candidates that have not been described before
(**Supplementary Table 3-4**), namely monocarboxylate transporter 4
(SLC16A3), filamin-A (FLNA), peroxisomal 3-ketoacyl-CoA thiolase (ACAA1),
transmembrane emp24 domain-containing protein 7 (TMED7) and
sarcoplasmic/endoplasmic reticulum calcium ATPase 1 (AT2A1)
(**Supplementary Table 3-4**). Reports of direct interactions between these

proteins and EGF are not available, but it was shown before that SLC16A3
co-locates with CD147 in breast cancer cells(Gallagher et al. 2007), which in
turn is associated with EGFR in similar lipid domains (Vial and McKeown-
Longo 2012) suggesting that SLC16A3 resides in the neighbourhood of
EGFR at the cell surface (Dai et al. 2013). [...] Given the experimental setup,
the candidates identified from HATRIC-LRC experiments can generally be the
result of four scenarios (1) there is a direct interaction of the ligand with the
target receptor; (2) the protein is in close proximity of the target receptor
(“neighbourhood protein”); (3) the protein gets upregulated in response to
treatment with the ligand and gets overrepresented in the background binding
of HATRIC (e. g. we use approximately 8 times more EGF than is used for
stimulation experiments) or (4) the identified candidate is a false positive. A
single HATRIC-LRC experiment does not allow us to delineate which type of
interaction was observed, but the validation experiments and the cited data
clearly underline the biological relevance of the identified proteins. The
analysis pipeline was optimized to allow for the identification and ranking of
receptor candidates. However, the resulting data have to be analyzed
carefully and more stringent receptor spaces can be defined based on the
identification of positive control receptors or the ligand (e.g. EGF). Identified
candidates need validation in tailor-made follow-up experiments, such as
siRNA-based approaches. [...]

7. For the viral work, an interesting follow-on functional study is shown. For this work,
the authors should show the level of depletion achieved by the siRNAs for each of these
targets. For interpretation of this data, it is important for the reader to know if all of the
candidate receptors were successfully depleted and, if so, by how much?

• We would like to thank the reviewer for the comment and we have addressed
this now in our revised manuscript. To this end, we performed real time RT-
PCR for all 21 genes and quantified the gene depletion level (**Pbp Figure**
**10/Supplementary Figure 8**). The experiment was repeated twice with
similar results. Twenty genes showed above 70% depletion of the respective
mRNA i.e. >90%, 9 genes; >80%, 6 genes; >70%, 5 genes. A single gene,
CRTAP, showed no reduction upon siRNA treatment. We conclude that IAV
infection in CRTAP siRNA-treated cells were reduced to unknown off-target
effects (see original manuscript (**Fig. 2F**)). Thus, we removed CRTAP from
the infection data figure (**Fig. 2F**).

**PbP Figure 10 / Supplementary Figure 8** | Results of qPCR from siRNA-transfected cells.

siRNA-mediated silencing of IAV-interacting candidates was assessed using a $\Delta\Delta C_t$ method

to determine the relative gene expression from qPCR data using HPRT as the housekeeping

gene. For all genes tested, siRNA-mediated knockdown resulted in 70-98% reduction in

mRNA levels compared to non-targeting siRNA control. The bars represent relative gene

expression relative to the control taken from biological duplicates with standard deviation.

The experiment was repeated twice with similar results.

**PbP Figure 11 / Figure 2F** | Effect of siRNA-mediated depletion of candidate receptors on
 IAV infection of A549 cells. Experiments were conducted in triplicate. Infection scores from
 siRNA-treated samples were normalized to control samples transfected with non-targeting
 siRNA (shown in grey). Data are presented as boxplots with whiskers from minimum to
 maximum values.

- • **Changes to the manuscript:** [...] To determine whether candidate receptors
 impact IAV entry, we depleted A549 cells of 21 of these proteins using short
 interfering RNA (siRNA) and analyzed infection efficiency. siRNA-mediated
 depletion of more than 70% was confirmed by real time RT-PCR in 20 genes.
 We excluded cartilage-associated protein (CRTAP) from further analysis as
 siRNA treatment failed to deplete it (**Supplementary Fig. 8**). Depletion of four
 proteins, phospholipase D3 (PLD3), ribophorin I (RPN1), folate transporter 1
 (SLC19A1) and vesicular integral-membrane protein VIP36 (LMAN2) reduced
 IAV infection by more than 50% relative to cells treated with control siRNA
 (**Fig. 2f**). [...]

8. Viral mediated entry is a very complicated cellular process, utilizing a wide variety of
 physiological pathways. I wonder if 21 randomly selected reasonably high abundance cell-
 surface expressed proteins were chosen for this experiment, would the 'hit rate' would be
 lower than what was seen here? So many proteins would affect one aspect or another of
 viral entry...

• We believe the hit rate would be considerably lower if random cell surface
proteins were selected. The reason is the following: Of the validated genome-
wide siRNA screens performed against IAV infection (Brass et al. 2009)
(Karlas et al. 2010) (König et al. 2010), the number of targeted genes were
17877, 22843, and 19628, respectively, of which ‘validated hits’ (hit genes
against which depletion of the gene was confirmed by at least two siRNAs)
were only 129 (0.72%), 168 (0.73%), and 219 (1.1%), respectively. In our
HATRIC-LRC screen, we retrieved 21 genes from which we removed one
gene (CRTAP) due to failed siRNA depletion. Of the remaining 20, 2 genes
(RPN1, PLD3) reduced infection >80% (strong decreaser hits), another 2
genes (SLC19A1, LMAN2) reduced infection >55% (weak decreaser hits),
and another 2 genes (APMAP, CLPTM1) increased infection >70% (increaser
hits). The depletion of these genes was verified by RT-PCR. It is clear from
this result that the genes enriched using the HATRIC-LRC approach were
highly enriched in hit genes (20% i.e. 4 out of 20) compared to a randomly
selected pool of genes (Please also find more comments on pages 28 and
following of our point-by-point response). 14 out of the 20 genes did not give
a noteworthy effect on IAV infection when knocked-down as single genes.
However, that silencing of a single factor did not completely attenuate
infection was not surprising. This likely reflects the complex nature of
influenza-host cell interactions in which multiple virus and cellular factors
each contribute to successful and potentially cooperative binding and
infection.

Minor comments:

1. The information provided on the MS results is minimal. While it is great that the MS
raw files have been made available, some minimal information should be provided in the
manuscript/supplementary info. For example, no peptide-level results are shown or provided.
At a minimum, the number of unique peptides identified/quantified for each protein should be
provided in the manuscript. Ideally, some information on the quantitative variability seen
between different peptides from the same protein should also be provided.

• We added tables containing the complete information on peptides used for
quantification for each data set (Progenesis output tables, **Supplementary**
**tables 1A, 4A, 5A, 6A, 7A, 9A**) and the outcome of our statistical analysis
containing all information necessary to create volcano plots (**Supplementary**
**tables 1B, 4B, 5B, 6B, 7B, 9B**). This information will provide a transparent
overview on the quality of the data.

2. More details on the statistical methods used would be helpful. How were protein-level p-values determined? How was quantitative data from individual peptides combined? How were the different technical replicates used for this calculation? What modules from MSstats were used?

- Thank you for noticing, it was indeed very short and we rectified it now.
- **Changes to the manuscript:** For label-free quantification, proteins were filtered for cell surface location, based on the cell surface protein atlas (Bausch-Fluck et al. 2015) and the human surfaceome (Omasits U. et al., manuscript in preparation; **supplementary table 3**) and non-conflicting peptide feature intensities extracted with Progenesis QI (Nonlinear Dynamics). The output of Progenesis is a list of quantified spectral features representing peptides of cell surface proteins with multiple charge states and differential modifications. In MSstats3 (v3.2.2), the features were log-transformed, and then subjected to constant normalization (Choi et al. 2014). Protein fold changes and their statistical significance between paired conditions were tested using at least two fully tryptic peptides per protein or one fully tryptic peptide per protein for the 1 million cell experiment. The minimum intensity for each peptide feature was set to 500. Tests for significant changes in protein abundance across conditions are based on a family of linear mixed-effects models. In the last step of the analysis, *P* values are adjusted for multiple comparisons to control the experiment-wide FDR at a desired level using the Benjamini-Hochberg method. Proteins were considered candidates if they showed a fold-change of 1.5 or higher and an adjusted p-value of 0.05 or lower.

3. Methods section: pH required for digestion buffer description.

- **Changes to the manuscript:** Cells were pelleted, washed twice with PBS (pH 7.4) to remove unbound HATRIC, and lysed with 8M Urea, 0.1% RapiGest SF (Waters) containing protease inhibitors (cOmplete, Roche), pH 8.

4. The main body text describing the 1 million cell experiment should mention that this was in MDA-231 cells.

- **Changes to the manuscript:** [...] From as little as one million MDA-MB-231 cells per sample, we were able to unambiguously identify EGFR as the

636 receptor for HATRIC-coupled anti-EGFR antibody and transferrin receptor
protein 1 (TFR1) as the receptor for HATRIC-coupled Holo-transferrin (TRFE)
(Fig. 2b) which was not possible with TRICEPS-LRC (Supplementary
Figure 5, Supplementary Table 6). Where possible, we recommend the
usage of 5-20 million cells in order to detect low copy number receptors
based on a given sensitivity of the MS instrument used for analysis.[...]

Reviewer #2 (Remarks to the Author):

The manuscript by Sobotzki et al. describes a novel technique to fish for cellular receptors
for a variety of ligands, including proteins, small molecules and viruses. The authors
developed a trifunctional organic compound, which can covalently link proteinaceous ligands
and through a second reactive group covalently link receptor molecules after incubation of
the compound-ligand molecule with target cells. Finally a third reactive group allows the
purification of putative receptor-ligand-compound complexes by click chemistry. Purified
proteins are quantified by LC-MS/MS analysis using standard protocols and compared to
control conditions, in which receptor binding of the compound labeled ligand is competed for
with an excess of unlabeled ligand or the compound is rendered inactive by glycine
quenching. The authors perform four proof-of-principle experiments. First they use the
method to confirm epidermal growth factor (EGF) binding to EGF receptor (EGFR). Second,
they determine the experimental threshold using transferrin binding to its cognate receptor.
Third, the authors demonstrate applicability to the small organic ligand folate. Lastly, they
perform an experiment with influenza A virus bound to human lung epithelial cells. While the
compound synthesis and the first three proof-of-principle experiments are well designed and
controlled, the experiments on influenza virus require some attention. Moreover I
recommend a thorough discussion of the discrepancy between the identified IAV attachment
factors and previously described host factors (multiple RNAi screens). Also the false positive
rate should be discussed as detailed below. Overall, the manuscript is however very well
written and the description of methods is clear to non-expert readers. Statistical analysis of
MS data is sound. Once the points below are addressed, I favor publication of this
description of an exciting and promising new technology, which is clearly of interest to
various fields of biology.

- • We would like to thank the reviewer for the insightful summary. We have
revised the manuscript to include a section that clearly discusses the role of
the identified IAV entry facilitators or inhibitors and what was previously
known about these proteins.

Major comments:

- 1. Fig. 2e. Why was insulin used as control ligand? While the first three experiments
were well controlled, this control seems random. New datasets with compound-free virus
competition or quenched virus would seem better controls.

• This is a valid and appreciated argument raised from this reviewer and we
agree with the reviewer that on the first glance, the choice of this ligand
appears random. However, we would like to politely point out, that we
deliberately chose insulin as a technical control ligand in the virus-receptor
capture experiment. In contrast to the other experiments reported in the
paper, we didn't know which receptors to expect for influenza. Given the
rather long protocol and the risk of bias in the result due to differential sample
processing, we wanted to use a ligand with known receptor specificity that
would allow us to come to a distinct decision if the experiment was successful
on the technical level and if the results qualify for follow-on experiments.
However, we do agree with the reviewer that the best experimental setup is to
have three samples tested in parallel: A ligand with known specificity (positive
control), the virus (the sample) as well as competition with unmodified virus or
quenched virus (negative control). For future experiments, this expanded
setup might lead to improved scoring of candidates and could be beneficial
for receptor identification.

2. Fig. 2e. Compound labeling of viruses can strongly affect infectivity. The authors
should perform control experiments, in which they compare titers of virus before and after
labeling. Moreover the effect of labeling on the specific infectivity (infectious particle /
genome copy number) should be measured.

• Please see a combined response below.

3. Fig. 2e. Compound labeling of small enveloped viruses such as influenza A virus
may affect its entry route. The authors should experimentally demonstrate that the entry
pathway into A549 cells is not altered after compound labeling of the virus particles using
inhibitory compounds and/or imaging techniques.

• The response is combined for the above two points: We thank the reviewer
for these comments. It is indeed possible that compound labeling with
HATRIC (albeit at 2 HATRIC molecules per virion) could affect infectivity and
could alter the entry pathway of IAV particles. We performed IAV endocytosis,
primary infection, and multi-step growth assays using IAV labeled with
HATRIC or incubated with buffer alone (**PbP Figure 12 / Supplementary
Figure 6**). HATRIC-conjugated or unconjugated IAV particles were processed
for the endocytosis and infection assays as described previously (Banerjee et
al. 2011). For IAV titration, virus was infected in 10-fold serial dilutions onto

MDCK II cells. Plaques were counted after 3 days of infection and the viral
 plaque forming unit (PFU) was calculated per mL of inoculant.
 • There was an approximately 30% decrease of IAV endocytic uptake
 (Banerjee et al. 2011), infection, and replication titer when viruses were
 conjugated to HATRIC (**PbP Figure 12 / Supplementary Figure 6**). This
 suggests that HATRIC coupling reduces IAV endocytosis, but those that have
 been endocytosed, infect and replicate as well as non-conjugated virus.

**PbP Figure 12 / Supplementary Figure 6** | Impact of HATRIC-coupling to influenza on
 efficiency of viral endocytosis (A), infectivity (B) and IAV titer (C). IAV particles were left
 unchanged (control) or coupled to HATRIC (HATRIC) and submitted to endocytosis assay
 (25 min post warming) and infection assay (7 hpi) as described previously (Banerjee et al.
 2011). For IAV titration, control and HATRIC-conjugated virus was infected in a 10-fold serial
 dilution series onto a monolayer of MDCK II cells and overlaid with 1.2 % Avicel containing
 MEM. Plaques were counted after 3 days of infection and the plaque forming unit (PFU) was
 calculated per mL of inoculant.

 • IAV endocytosis utilises two main pathways i.e. clathrin-mediated
 endocytosis, macropinocytosis, and a third, poorly characterised pathway that
 is dynamin-independent and actin-dependent. To confirm that HATRIC
 coupling did not influence the endocytic pathways used by IAV, we performed
 endocytosis assays using inhibitors against dynamin (Dyngo-4a) and
 micropinocytosis/fluid uptake (EIPA). We normalised the decrease in
 endocytic uptake compared to the DMSO-treated cells for the control and
 HATRIC-coupled IAV, respectively (**PbP Figure 13 / Supplementary Figure 7**).
 Based on the inhibitory effects of Dyngo-4a and Dyngo-4a/EIPA
 combined, we conclude that the endocytic pathways used for virus cell entry

are identical for both HATRIC-treated and non-treated IAV. EIPA treatment alone did not reduce IAV endocytosis (not shown). HATRIC did not influence virus attachment to the cell surface.

PbP Figure 13 / Supplementary Figure 7 | A549 cells were pretreated with Dyngo-4a (50µM) or both Dyngo-4a and EIPA (80 µM) for 30 min, after which equal volumes of IAV were bound for 45 min on ice in the presence of the drug(s). The cells were then washed and incubated at 37°C for 25 min in the presence of the drug(s), fixed and stained for endocytosis analysis (Banerjee et al. 2011).

- **Changes to the manuscript:** [...] We used HATRIC-LRC to shed light on the complex interactions between IAV and its host cells. Human IAV H3N2 (strain X-31) was coupled to HATRIC and it was demonstrated that although coupling reduces IAV endocytosis, the particles used similar endocytic pathways to the wild-type virus (**Supplementary Figures 6 & 7**). We conducted H3N2-based HATRIC-LRC on to 20 million human lung adenocarcinoma (A549) cells and compared to the control ligand insulin. We identified 24 virus-interacting candidates (**Fig. 2e, Supplementary Table 7-8**). [...]

4. Fig. 2 and lines 316-331: The authors should explain the filtering for cell surface
molecules in the main manuscript, not only in the methods. They should disclaim, which
fraction of the identified proteins was cell surface associated according to e.g. GO
annotation.

- Please find a detailed response to this comment on page 6/7 of the point-by-point response.

5. Fig. 2e. Why was a nuclear pore protein (NUP210) identified despite the surfaceome filtering? What is the leakiness of the method towards cytoplasmic or nuclear proteins?

- As pointed out earlier, we filter our data for proteins that are annotated to be located at the cell surface. We identified NUP210 here because it is in this filter list, namely in the cell surface protein atlas (CSPA) filter list annotated as high confidence protein (Bausch-Fluck et al. 2015). NUP210 contains several N-glycosylation sites as well as transmembrane domains, allowing in theory the localization at the plasma membrane. We conducted confocal microscopy imaging and HATRIC co-localized with cell surface staining. These data show that HATRIC doesn't penetrate cells which allows us to exclude this as a technical contamination as a nonspecific protein. However, our and previous experiments provide evidence that NUP210 might be located at the cell surface at some point in its lifetime: Greber et al. found that Nup210 is a transmembrane nucleoporin with a long luminal domain, a single transmembrane segment, and a short 55 amino acid nuclear/cytoplasmic tail, a structure that resembles that of viral membrane fusion proteins (Greber, Senior, and Gerace 1990). Therefore, it might be possible that a fraction of Nup210 could function as a fusogenic protein at the plasma membrane. However, when fractionating postmitotic myotubes by sequential centrifugation, Nup210 was only detected in the nuclear fraction (D'Angelo et al. 2012).
- **Changes to the Manuscript:** [...] Evaluation of a number of aniline derivatives regarding their solubility, cytotoxicity and capability to enhance hydrazone formation between aldehydes on cell surface proteins and the HATRIC-hydrazide on living cells led to identification of 5-methoxyanthranilic acid (5-MA, **Fig. 1c**). 5-MA catalyzed hydrazone formation at a non-toxic concentration at pH 7.4 more efficiently than 2-amino-4,5-dimethoxy benzoic acid (ADA) (**Fig. 1c, Supplementary Fig. 1**). Additionally, replacing the original Trifluoroacetyl-protection group of TRICEPS by an acetone-derived protection group in HATRIC enabled higher yield of hydrazone formation on live cells (data not shown). Last, we confirmed that under the chosen conditions, HATRIC does not penetrate cells, to avoid contamination with intracellular proteins (**Supplementary Fig. 2**). [...]

**PbP Figure 14 / Supplementary Figure 2 | HATRIC co-localizes with cell surface**
**staining as shown by confocal microscopy imaging** (RED=HATRIC-Amine-Cy3,
GREEN=Sulfo-NHS-Cy5, BLUE=Hoechst). HATRIC was pre-coupled to equimolar amine-
Cy3 (Lumiprobe) in 25mM HEPES (pH 8.2) for 1.5h at RT and 300rpm in the dark. MDA-MB-
231 cells cultured on coverslips were oxidized with 1ml 1.5mM sodium periodate in PBS, pH
6.5 for 15min and labeled with 6μM HATRIC-Cy3 or amine-Cy3 (Control w/o HATRIC) in 1ml
PBS with 5mM 5-MA (pH 7.4) for 1.5h at 4°C shaking in the dark. As a cell surface marker,
cells were labeled with 0.5ml 1mM sulfo-NHS-Cy5 (Lumiprobe). Nuclei were stained with
0.5ml 1μg/ml Hoechst (Molecular probes H1399) for 10min at 4°C. Cells were fixed with 4%
paraformaldehyde for 10min at RT, mounted with anti-fade mounting medium (Molecular
Probes Prolong Gold Antifade reagent P36934) and analysed by confocal microscopy (Leica
TCS SP2). For a permeabilized control, cells were first stained with sulfo-NHS-Cy5 and
fixed, and then permeabilized with 0.1% Triton X-100 for 10min at RT, before oxidation and
labeling with HATRIC-Cy3.

6. Line 177: Multiple RNA interference (RNAi) screens on influenza have been
published, with some overlap. It is recommended that the authors discuss in more detail why
on the one hand the published RNAi hits were not discovered in their HATRIC experiment
and on the other hand, why their MS hits were vice versa not previously identified in any of
the influenza host factor searches.

● Of the 5 independent siRNA screens on IAV that were published, three have
 been validated (Brass et al. 2009; Karlas et al. 2010; König et al. 2010). Of
 the 129, 168, 219 genes that were validated as hits from these three screens,
 34 genes were shared in two or more of them. Only 3 genes (ARCN1,
 ATP6AP1, and COPG) were shared among all three. The little overlap
 between individual IAV RNAi screens has been described elsewhere (Stertz
 and Shaw 2011). This low number of overlapping genes is similar to what has
 been observed for the RNAi screens performed against HIV-1. We compared
 our top 7 decreaser hits (RPN1, PLD3, SLC19A1, LMAN2, ASPH, CD151,
 RPN2) with the 34 genes above using STRING: functional protein association
 networks (PbP Figure 15).

**PbP Figure 15 | Network of influenza interaction candidates.** We compared our influenza
 entry candidates to 34 genes that were validated as hits in at least two of the published,
 independent IAV siRNA screens.

- ● RPN1 (Ribophorin I), RPN2 (Ribophorin II), PLD3 (Phospholipase D3), ASPH
 (Aspartate beta-hydroxylase) formed connections with the 34 genes. Below
 are the genes that have been published as hits for IAV which are similar to
 function to PLD3 and SLC19A1 [Solute carrier family 19 (folate transporter),
 member 1] derived via HATRIC-LRC.
 - ○ PLD2 (Phospholipase D2) (*Karlas*) (also see (Oguin et al. 2014)).
 - ○ Solute carriers:
 - ■ SLC4A3, SLC2A2, SLC1A3, SLC35A1 (*Brass*)

- ■ SLC22A6 (*Karlas*)
- ■ SLC48A1, SLC6A19 (*König*).
- ● However, IAV X31 expresses the external genes derived from a H3N2
- influenza A strain, thus *may use a different subset of cell surface genes to*
- *enter cells* compared to PR8 and WSN (H1N1). Below is a summary of the
- IAV strains used in our HATRIC-LRC screen and other published validated
- screens (**PbP Figure 16**):
- ○ Sobotzki et al.: X31(reassortant strain of external genes of
- [A/Aichi/2/68 (*H3N2*) and internal genes of A/PR/8/34 (H1N1)];
- ○ Brass et al (2009): A/PR/8/34 (H1N1);
- ○ Karlas et al., (2010): A/WSN/1933 (H1N1);
- ○ König et al., (2010): recombinant A/WSN/1933 (H1N1) in which the
- HA gene was replaced by *Renilla* luciferase.

IAV RNAi screens	Sobotzki et al.	Brass et al. (2009)	Karlas et al. (2010)	König et al. (2010)
Host cell	A549 (Human)	U2OS (Human)	A549 (Human)	A549 (Human)
Virus	X31	PR8	WSN	recombinant WSN
Readout	NP expression	HA expression	1 st cycle: NP expression 2 nd cycle: luciferase activity	Luciferase activity
siRNA source	Dharmacon	Dharmacon	Qiagen	Qiagen
Length of RNAi treatment	72 h	72 h	48 h	48 h
Time of assay readout	7 hpi	12 hpi	24 hpi	12, 24, 36 hpi
Virus stages captured	Attachment until protein expression	Attachment until HA surface trafficking	Attachment until budding/release	Attachment until protein expression
Genes targeted	20	17,877	22,843	19,628
Validated hits	4	129	168	219
Hit rate	20%	0.72%	0.73%	1.1%

**PbP Figure 16** | Descriptions of the validated siRNA screens performed with
influenza virus. Adapted from Stertz & Shaw, (2011). hpi, hours post infection.

- ● **Changes to the manuscript:** [...] Of the receptor candidates identified using
- HATRIC-LCR, none have been implicated previously in mediating H3N2
- infection. However, it has been shown that related phospholipase γ 1 (PLC- γ 1)

signaling is activated by H1N1 and mediates efficient viral entry in human
epithelial cells(Zhu, Ly, and Liang 2013). Of the 5 independent genome-wide
siRNA screens on IAV that were published, three have been validated (Brass
et al. 2009; Karlas et al. 2010; König et al. 2010). Of the 129, 168, 219 genes
that were validated as hits from these three screens, 34 were shared in two or
more. Only 3 genes (ARCN1, ATP6AP1, and COPG) were shared among all
three. A comparison of our top 7 decrease genes (RPN1, PLD3, SLC19A1,
LMAN2, ASPH, CD151, RPN2) with the 34 genes revealed mild functional
overlap as shown by STRING (Supplementary Fig. 9). We also had 4 strong
hits (i.e. increased or decreased infection by more than 70%) out of 20
validated genes – a hit rate of 20% - which is considerably higher compared
to the genome-wide screens (~1%). [...]

7. A differentiated discussion on the limitations of the technology is missing. Can any
small ligand be linked to the HATRIC compound without affecting receptor affinity? What are
the requirements of organic compounds to be successfully fused to HATRIC by synthesis?

• Like every other technology HATRIC-LRC does have limitations: (1) HATRIC-
LRC is a screening technology that may lead to identification of candidate(s)
which need to be further validated in order to investigate the precise role of
the identified receptor in the biology and signaling of the ligand. (2)
Identification of “nonspecific” proteins makes data filtering indispensable.
Data filtering might lead to exclusion of proteins that are relevant candidates
but are not included in the filter list. (3) HATRIC-LRC can be coupled to (small
molecule) ligands that bear primary amine groups (no other prerequisites
required) which might require a more complex synthesis strategy and (4)
modification of small molecules with a rather large compound like HATRIC
(Mw 1171.4 g/mol) may drastically change activity of the compound.
However, there is no other method to investigate receptor binding of ligands
on live cells requiring this little amount of cells and there is no other method
that allows for direct identification of cell surface receptors for small
molecules. Also, identification of false positives may also occur with all other
available screening approaches, such as TRICEPS-LRC and ASB.

• **Changes to the manuscript:** [...] We demonstrated that HATRIC-LRC
enables ligand-receptor identification from as few as 1 million cells at
physiological pH through new chemistry combining HATRIC, a water-soluble
catalyst, and click chemistry-based protein-level affinity purification in a
competition-based workflow. Even though HATRIC-LRC is a screening

technology that leads to candidate receptors, including potentially false
positive receptor candidates, which need to further validated, its ability to
detect biologically meaningful ligand-receptor interactions remains
unmatched. The power of HATRIC-LRC to detect functionally relevant cell
surface interactions was demonstrated using ligands ranging from small
molecules to intact influenza A virus particles. [...]

8. The full MS datasets should be disclosed in supplementary tables and deposited in
public online repositories such as the EMBL/EBI IntAct database. In particular for the
influenza A virus experiment.

• All MS data have been deposited to the MassIVE repository
(<http://massive.ucsd.edu/> MassIVE ID: MSV000081228).

Minor comments:

1. Full protein names are not mentioned. Please write out the full names at first
mentioning of a protein abbreviation, such as FOLR1.

• Thank you for noticing, this has been rectified.

2. Supplementary table 3: The human surfaceome should be presented with separate
columns for gene name, protein name and Uniprot accession number for easier accessibility.

• Thank you for proposing this, we adapted the list accordingly. Please note
that the entry Q5VU13 became obsolete.

3. Fig. 2e,f. The gene/protein names do not match between Fig. 2e, Fig. 2f, Tab. S2
and Tab S4. If the authors decide to use protein names in Fig. 2e and gene names in Fig. 2f,
it is advisable to include both – protein names and gene names – in Tab S2 and S4 to allow
the reader to match the datasets.

• Thank you for noticing, this was rectified accordingly. The table S2 was
changed to match the figures (all proteins are reported with their gene names
now): MRP4 was changed to ABCC4; CALX was changed to CANX; CBPM
was changed to CPM; PO210 was changed to NUP210; UGGG1 was
changed to UGGT1; TOIP2 was changed to TOIR1AIP2; MRP1 was changed
to ABCC1; TOIP1 was changed to TOIR1AIP1; S19A1 was changed to
SLC19A1; CLPT1 was changed to CLPTM1

4. Certain proteins, which were silenced (Fig. 2f), are not included in Tab. S2 or
annotated differently. Examples are SLC19A1, NUP210, ABCC4.

- • Thank you for noticing, this was rectified accordingly by adapting the gene
names as described above.

Reviewer #3 (Remarks to the Author):

In the manuscript from Sobotzki et al., the authors demonstrate their development of next-
generation LRC method. Having been the leading developers of the first-generation
reagents, TRICEPS-LRC, the Wollscheid laboratory is well-suited to evolve this useful
technology for improved coverage, applicability, and sensitivity. The updated methodology,
termed HATRIC, still employs the key step of receptor sugar alcohol to aldehyde periodate
oxidation, and subsequent coupling to the hydrazine-containing probe. However, the authors
optimized the periodate oxidation to achieve high efficiency at neutral pH. In addition, the
authors introduced Click chemistry in the HATRIC reagent. These optimizations directly
contribute to the improved sensitivity of the approach, with a minimum requirement of
between 1 -2 orders of magnitude less cellular material. The authors experimentally
demonstrated the results of HATRIC-LRC with 1 million cells, though as mentioned in the
comments below, the explanation of this experiment in the manuscript could be improved.
The work nicely demonstrates the broad application of the method to a range of ligands,
including the small molecule folate, the polypeptide EGF, and the intact virus, influenza A.
The authors convincingly demonstrated that their technology could identify biologically
relevant cell surface receptors of IAV by validation with siRNA knockdown of candidate IAV
cell surface receptors during infection. However, as mentioned in the main comments
section, the authors did not fully discuss why none of the known IAV receptors were
identified.

Overall, this is a strong methodological study with significant application to biomedical and
pharmaceutical research, particularly in contributing to the characterization of orphan
receptors. The authors do have a few outstanding and several minor points to address;
however, if these can be addressed, I would recommend the manuscript for publication.

Main Points

1. A general main point is the lack of discussion related to novel identified candidates or
lack of identification for known candidates in the case of IAV. For instance, in addition to
identifying the known receptors for the EGF and folate ligands, the authors found several
other putative candidates, which the authors did not discuss.

- • We would like to thank the reviewer for the valuable suggestion to add
information about putative receptor candidates for the ligands EGF and folate.
The lack of some details is mainly due to the initial space constraints of the
format. We now added more details in the text and in the supplementary
information.

- **Changes to the manuscript** about candidate receptors identified for EGF can be found on page 10 of the point-by-point response.
 - **Changes to the manuscript** about candidate receptors identified for folate: We incubated the folate-HATRIC conjugate with 20 million HeLa Kyoto cells at pH 7.4. In the control, we added six-fold excess of unmodified folate. We detected interactions with FOLR1 and with a small set of further receptor candidates (**Fig. 2c, d; Supplementary Table 6**). We suggest that other folate receptors (e. g. FOLR2) were not identified as their affinity towards folate is lower than the affinity of FOLR1, i. e. FOLR2 has a two-fold reduced affinity for folate compared to FOLR1 or because they are not expressed in HeLa Kyoto cells (Brigle et al. 1994). Related approaches studied methotrexate-based labeling of FOLR1, but western blot read-outs didn't provide information about other folate receptor candidates (Fujishima et al. 2012).

What percent were known or predicted cell surface or secreted proteins? In addition, for the
IAV experiments, the authors state: " We identified 24 virus-interacting candidates (Fig. 2e,
Supplementary Table 2)." Before discussing the siRNA results, the authors should expand
on their statement. Later in the manuscript, the authors mention that none have been
previously implicated. However, it might be appropriate for the authors to briefly discuss
here, (1) that these targets didn't include the known receptors, (2) how many known receptor
targets are there for IAV, (3) their thoughts on why HATRIC did not capture them?

- Thank you for your insightful request. We would like to politely point out that there are no confirmed receptors for the specific IAV strain that we used in the paper. Other studies in the influenza field are reviewed on p. 28 of the point-by-point response and following pages.

2. Did the authors evaluate intracellular generation of aldehydes with the improved
periodate oxidation using 5-MA? Is the HATRIC reagent cell permeable, e.g. with a small
molecule conjugate like folate?

- We would like to point out politely that 5-MA doesn't affect periodate oxidation (oxidation with sodium periodate is a separate step in the protocol), but catalyzes hydrazone formation between the acetone-protected hydrazone of HATRIC and cell surface aldehydes that were generated before through periodate oxidation. We acknowledge that the major issue here seems to be an unclear presentation of our proceedings and we rectified this in our revised manuscript. However, we agree with this reviewer that investigating cell

permeability of HATRIC is particularly interesting in the context of small
molecule-based capture experiments. We conducted confocal microscopy
imaging and HATRIC co-localized with cell surface staining. This shows that
HATRIC doesn't penetrate cells which avoids non specific labeling of
intracellular proteins. Please see the data and figure provided for the previous
reviewer on page 26/27 for more detailed information.

- • **Changes to the manuscript:**[...] First, the ligand is linked through a primary
amine to the NHS-moiety of HATRIC (**Fig. 1b**). Second, living cells are mildly
oxidized with sodium-meta-periodate to generate aldehydes from cell surface
carbohydrates. Third, the HATRIC-ligand conjugate is added to the cells in
the presence of catalyst 5-methoxyanthranilic acid (5-MA) and receptor-
capture performed at pH 7.4. The ligand enhances local HATRIC reactivity in
the vicinity of the target receptor or receptors, and receptor aldehydes react
with the acetone-derived hydrazone of HATRIC. In the control, the HATRIC-
conjugated ligand is applied to the cells in the presence of an excess
unmodified ligand. Here, the ligand-HATRIC conjugate reacts randomly with
cell surface glycoproteins. [...] Evaluation of a number of aniline derivatives
regarding their solubility, cytotoxicity and capability to enhance hydrazone
formation between aldehydes on cell surface proteins and the HATRIC-
hydrazide on living cells led to identification of 5-methoxyanthranilic acid (5-
MA, Fig. 1c). 5-MA catalyzed hydrazone formation at a non-toxic
concentration at pH 7.4 more efficiently than 2-amino-4,5-dimethoxy benzoic
acid (ADA). **Fig. 1c, Supplementary Fig. 1**). Additionally, replacing the
original Trifluoroacetyl-protecting group of TRICEPS by an acetone-derived
protection group in HATRIC enabled higher yield of hydrazone formation on
live cells (data not shown). Last, we confirmed that under the chosen
conditions, HATRIC does not penetrate cells avoiding contamination with
intracellular proteins (Supplementary Fig. 2). [...]

3. The overall strategy and figure panel (Fig 2b) to identify “EGFR as the receptor for
anti-EGFR antibody and transferrin receptor protein 1 (TFR1) as the receptor for Holo-
transferrin (TRFE) from 1 million cells per sample” is confusing. The idea of testing the limit
of detection for HATRIC (1 million cells) is clear, but how is this related to anti-EGFR
antibody? Is this used instead of HATRIC? What is the relationship between EGFR and
TRFE? This experiment should be described in the Methods section.

- The strategy of this experiment was to test if we were able to identify the receptors for well-known ligand-receptor pairs from as little cells as possible. To this end, we selected the ligand “anti-EGFR antibody” that we knew binds reliably to EGFR on the cell surface. We conducted a Standard HATRIC-LRC where HATRIC is coupled to this antibody (or to holo-transferrin in the control reaction) and successfully identified EGFR (or transferrin receptor protein 1) from 1 million cells. We acknowledge that this was presented in a suboptimal way in the main text and have adapted the manuscript accordingly below.
 - **Changes to the manuscript:** As HATRIC-LRC is based on protein-level purification, more than one peptide is commonly identified per protein, such as exemplified by EGFR (**Supplementary Fig. 4**). Therefore, we investigated the HATRIC-LRC detection limit with respect to the amount of starting material needed for successful receptor identification. From as little as one million MDA-MB-231 cells per sample, we were able to unambiguously identify EGFR as the receptor for HATRIC-coupled anti-EGFR antibody and transferrin receptor protein 1 (TFR1) as the receptor for HATRIC-coupled Holo-transferrin (TRFE) (**Fig. 2b**) which was not possible with TRICEPS-LRC (**Supplementary Figure 5, Supplementary Table 6**). Where possible, we recommend the usage of 5-20 million cells in order to detect low copy number receptors based on a given sensitivity of the MS instrument used for analysis.

Minor Points

1. The first description of HATRIC in Fig 1b, has an application that is targeted to
specific glycoproteins or glycoprotein classes using ligand coupling. Although the first
generation of TRICEPS was also a LRC method, could HATRIC (and in general these
technologies) be used to gain broad capture of the glycoproteome in the absence of ligand
coupling.

- Yes, in principle it is conceivable to use HATRIC to study the glycoproteome at the cell surface. In such a setup, we suggest to quench the amine-reactive NHS-moiety of HATRIC with glycine to avoid unwanted side reactions. However, technologies based on two-functional compounds were developed before addressing exactly that question, such as biocytin hydrazide-based cell surface capture which might be more suitable to address such questions (Wollscheid et al. 2009).

2. In general for LRC technologies, is ligand-receptor activation and receptor-mediated
events such as internalization an issue?

• Thank you for that insightful comment. We conduct the whole experiment on
ice which prevents such receptor-mediated internalization events as also can
be seen from our previously presented microscopy data.

• **Changes to the manuscript:** [...] First, the ligand is linked through a primary
amine to the NHS-moiety of HATRIC (Fig. 1b). Second, living cells are mildly
oxidized with sodium-meta-periodate to generate aldehydes from cell surface
carbohydrates. During the whole experiment, cells are kept on ice to prevent
any receptor-mediated internalization events. Third, the HATRIC-ligand
conjugate is added to the cells. The ligand enhances local HATRIC reactivity
in the vicinity of the target receptor or receptors, and receptor aldehydes react
with the acetone-derived hydrazone of HATRIC. In the control, the HATRIC-
conjugated ligand is applied to the cells in the presence of an excess
unmodified ligand. Here, the ligand-HATRIC conjugate reacts randomly with
cell surface glycoproteins. As alternative controls, HATRIC can be quenched
with glycine (negative control) or a ligand with known target receptors can be
employed as a positive control (not depicted in figure).

3. The authors state: “The novel workflow renders HATRIC-LRC independent of the
PNGase F deglycosylation reaction, ultimately enabling a more robust relative quantification
of cell surface receptors than is possible with first-generation LRC”. This seems to imply that
the first-generation LRC (assume TRICEPS-LRC) could not be performed without PNGaseF.
If TRICEPS-peptide capture was performed (as in the authors previous work), then I would
agree. However, couldn't TRICEPS-LRC be performed with a protein capture, as described
for HATRIC, which would allow bead-based digestion as well?

• It is in theory conceivable to conduct a protein-level capture with TRICEPS-
LRC, but as TRICEPS-LRC is based on biotin-streptavidin affinity purification.
Tryptic digestion on streptavidin beads will lead to major contamination with
streptavidin peptides and will lead to ion suppression during MS
measurements. These limitations are overcome with click chemistry-based
affinity enrichment in HATRIC-LRC.

4. Conceptual flow of Figure 1b needs improvement. In the text, the description of steps
follows from (1) periodate oxidation to (2) addition of HATRIC-LRC, but in Fig 1b, the
periodate step is not explicit until the second box, which is after HATRIC-LRC/arrow graphic.

The authors should illustrate the periodate oxidation step and resulting modifications
explicitly, before addition of HATRIC-LRC?

• Thank you for this remark, the reviewer is completely right. The oxidation is a
separate step that needs to be completed prior to adding HATRIC. We added
the oxidation step as a separate step to the figure now and hope it makes the
methodology easier to understand.

• Changes to the manuscript:

**PbP Figure 16 / Figure 1B** | Workflow of HATRIC-LRC for identification of target receptors
of ligands on live cells. First live cells are mildly oxidized with 1.5mM NaIO₄. HATRIC,
conjugated to the ligand of interest, is added to the oxidized cells. The ligand selectively
directs HATRIC to its glycoprotein target receptor where HATRIC reacts to generate azide-
tagged cell-surface glycoproteins catalyzed by 5-MA. In order to identify target receptors of
orphan ligands, a dual track experimental setup is employed. In the control, the HATRIC-
conjugated ligand is applied to the cells in the presence of an excess unmodified ligand.
Alternatively, HATRIC can be quenched with glycine for a negative control or a ligand with
known target receptors can be employed as a positive control (not depicted in figure). After
lysis and affinity purification of azide-tagged proteins with unbound proteins removed by
harsh washing, peptides are proteolyzed with trypsin. Peptides are identified with high-
accuracy mass spectrometry in a data-dependent acquisition mode followed by quantitative
comparison of peptide fractions from experiment and control to reveal specific enrichment of
candidate cell surface receptors. Target receptors are defined as proteins that have a fold
change of greater than 1.5 compared to the control as well as an FDR-adjusted p-value
equal to or smaller than 0.05, corresponding to a target receptor window in the volcano plot
that is framed by dotted lines and highlighted in red.

5. The authors could consider integration the chemical structure of the catalyst 5-methoxyanthranilic acid (Fig 1c) into Fig 1d, perhaps as a mini-graphic next to the dashed trace, or alternatively, into the supplement.

- Thank you for noticing, this was adapted accordingly.

PbP Figure 9 / Figure 1C | Flow cytometry traces of U-2932 cells incubated with HATRIC conjugated to dibenzocyclooctyne-Alexa Fluor 488 (DIBO-AF488) at pH 6.5 or pH 7.4 in the presence or absence of organocatalyst 5-methoxyanthranilic acid (5-MA) (Structure shown, Mw = 167.16 g/mol) or 2-amino-4,5-dimethoxy benzoic acid (ADA). HATRIC was quenched with glycine (Gly-) to avoid potential reaction of HATRIC's NHS-ester with aminogroups at the cell surface. Shift to the right indicates more efficient labeling with HATRIC-DIBO-AF488.

6. In volcano plots for Fig 2, since there are a limited number of significant candidates, the authors should consider labeling all points with gene symbols/arrows, as needed.

- Thank you for this suggestion, we updated Fig. 2A and Fig. 2C accordingly and it makes the plots easier to interpret. However, in Fig. 2E, we added only a number of gene names as the plot is comparably crowded.

7. For the IAV experiment, what was the rationale for choosing insulin as a control instead of quenched HATRIC? I assume this was a positive control? If so, this should be explained more explicitly. Given the authors employ several options for controls, a few sentences clarifying the practical selection of controls could be helpful, especially regarding the above two options. For instance, if the positive control and experimental condition share a receptor, then the ratio would be 1:1 and eliminated from consideration.

- This is a valid and appreciated argument raised from the reviewer and we agree with the reviewer that on the first glance, the choice of this ligand

appears random. However, we would like to politely point out, that we
deliberately chose insulin as a control ligand in the virus-receptor capture
experiment. Quite frankly, this was one of the first experiments where we
successfully conducted HATRIC-LRC and we didn't know about the
alternative control experiments. However, in contrast to the other experiments
reported in the paper, we didn't know which receptors to expect for influenza.
Given the rather long protocol and the risk of bias in the result due to
differential sample processing, we wanted to use a ligand with known
receptor specificity that would allow us to come to a distinct decision if the
experiment was successful and if the results qualify for follow-on experiments.
However, we do agree with the reviewer that the best experimental setup is to
have three samples tested in parallel: A ligand with known specificity (positive
control), the virus (the sample) as well as competition with unmodified virus or
quenched virus (negative control). For future experiments, this setup might
lead to different scoring of candidates and can provide valuable insights.

8. In Figure 2f, what is an infection score? If it has units, it should be defined in the
legend.

• The percentage of cells that are positive for IAV gene expression
(nucleoprotein, NP) was calculated. The average value of infection (%) in the
non-targeting siRNA-treated cells is normalised as an infection score of 1.0.

9. Include units of concentration on the x-axis in Supplementary Fig 1.

• Thanks for noticing, we updated the entire figure with additional data and also
updated the x-axis accordingly.

 **PbP Figure 8 / Supplementary Figure 1** | Cytotoxicity of aniline and aniline-derived
 organocatalysts on MDA-MB 231. MDA-MB 231 cells (20.000 cells/well in a 96-well plate)
 were treated with the indicated concentrations of catalyst in DMEM (pH adjusted to 7.4, 1%
 Pen/Strep) for 1.5h at 37°C. Supernatant was replaced by 100ul DMEM with 10%
 alamarBlue™ reagent (ThermoScientific) and incubated for 5h at 37°C in the dark. Assay
 was read out by a fluoreader (Ex: 545nm, Em: 590nm, automatic gain).

 10. In the Tables, the authors should check their gene names for accuracy. For instance,
 in Table S1, the entries P09110 and O15427, the genes listed do not match the UniProt
 annotated genes.

- • Thank you for noticing, this was rectified.

References for the point-by-point response

 Banerjee, I., I. F. Sbalzarini, P. Horvath, and A. Helenius. 2011. "Histone Deacetylase 8 Is
 Required for Centrosome Cohesion and Influenza A Virus Entry." *PLoS*.
 journals.plos.org.
 <http://journals.plos.org/plospathogens/article?id=10.1371/journal.ppat.1002316>.
 Bausch-Fluck, Damaris, Andreas Hofmann, Thomas Bock, Andreas P. Frei, Ferdinando
 Cerciello, Andrea Jacobs, Hansjoerg Moest, et al. 2015. "A Mass Spectrometric-Derived
 Cell Surface Protein Atlas." *PLoS One* 10 (3):e0121314.
 Bhat, Venugopal T., Anne M. Caniard, Torsten Luksch, Ruth Brenk, Dominic J. Campopiano,
 and Michael F. Greaney. 2010. "Nucleophilic Catalysis of Acylhydrazone Equilibration
 for Protein-Directed Dynamic Covalent Chemistry." *Nature Chemistry* 2 (6):490–97.
 Brass, Abraham L., I-Chueh Huang, Yair Benita, Sinu P. John, Manoj N. Krishnan, Eric M.

Feeley, Bethany J. Ryan, et al. 2009. "The IFITM Proteins Mediate Cellular Resistance
to Influenza A H1N1 Virus, West Nile Virus, and Dengue Virus." *Cell* 139 (7):1243–54.

Brigle, K. E., M. J. Spinella, E. H. Westin, and I. D. Goldman. 1994. "Increased Expression
and Characterization of Two Distinct Folate Binding Proteins in Murine Erythroleukemia
Cells." *Biochemical Pharmacology* 47 (2):337–45.

Choi, Meena, Ching-Yun Chang, Timothy Clough, Daniel Broudy, Trevor Killeen, Brendan
MacLean, and Olga Vitek. 2014. "MSstats: An R Package for Statistical Analysis of
Quantitative Mass Spectrometry-Based Proteomic Experiments." *Bioinformatics* 30
(17):2524–26.

Crisalli, Pete, and Eric T. Kool. 2013. "Water-Soluble Organocatalysts for Hydrazone and
Oxime Formation." *The Journal of Organic Chemistry* 78 (3):1184–89.

Dai, Lu, Maria C. Guinea, Mark G. Slomiany, Momka Bratoeva, G. Daniel Grass, Lauren B.
Tolliver, Bernard L. Maria, and Bryan P. Toole. 2013. "CD147-Dependent Heterogeneity
in Malignant and Chemoresistant Properties of Cancer Cells." *The American Journal of
Pathology* 182 (2):577–85.

D'Angelo, Maximiliano A., J. Sebastian Gomez-Cavazos, Arianna Mei, Daniel H. Lackner,
and Martin W. Hetzer. 2012. "A Change in Nuclear Pore Complex Composition
Regulates Cell Differentiation." *Developmental Cell* 22 (2):446–58.

Dirksen, Anouk, and Philip E. Dawson. 2008. "Rapid Oxime and Hydrazone Ligations with
Aromatic Aldehydes for Biomolecular Labeling." *Bioconjugate Chemistry* 19 (12):2543–
48.

Frei, Andreas P., Ock-Youm Jeon, Samuel Kilcher, Hansjoerg Moest, Lisa M. Henning,
Christian Jost, Andreas Plückthun, et al. 2012. "Direct Identification of Ligand-Receptor
Interactions on Living Cells and Tissues." *Nature Biotechnology* 30 (10):997–1001.

Frei, Andreas P., Hansjoerg Moest, Karel Novy, and Bernd Wollscheid. 2013. "Ligand-Based
Receptor Identification on Living Cells and Tissues Using TRICEPS." *Nature Protocols* 8
(7):1321–36.

Fujishima, Sho-Hei, Ryosuke Yasui, Takayuki Miki, Akio Ojida, and Itaru Hamachi. 2012.
"Ligand-Directed Acyl Imidazole Chemistry for Labeling of Membrane-Bound Proteins
on Live Cells." *Journal of the American Chemical Society* 134 (9):3961–64.

Gallagher, Shannon M., John J. Castorino, Dian Wang, and Nancy J. Philp. 2007.
"Monocarboxylate Transporter 4 Regulates Maturation and Trafficking of CD147 to the
Plasma Membrane in the Metastatic Breast Cancer Cell Line MDA-MB-231." *Cancer
Research* 67 (9):4182–89.

Greber, U. F., A. Senior, and L. Gerace. 1990. "A Major Glycoprotein of the Nuclear Pore
Complex Is a Membrane-Spanning Polypeptide with a Large Luminal Domain and a
Small Cytoplasmic Tail." *The EMBO Journal* 9 (5):1495–1502.

Karlas, Alexander, Nikolaus Machuy, Yujin Shin, Klaus-Peter Pleissner, Anita Artarini,
Dagmar Heuer, Daniel Becker, et al. 2010. "Genome-Wide RNAi Screen Identifies
Human Host Factors Crucial for Influenza Virus Replication." *Nature* 463 (7282):818–22.

Khan, M. F., X. Wu, P. J. Boor, and G. A. Ansari. 1999. "Oxidative Modification of Lipids and
Proteins in Aniline-Induced Splenic Toxicity." *Toxicological Sciences: An Official Journal
of the Society of Toxicology* 48 (1):134–40.

König, Renate, Silke Stertz, Yingyao Zhou, Atsushi Inoue, H-Heinrich Hoffmann, Suchita
Bhattacharyya, Judith G. Alamares, et al. 2010. "Human Host Factors Required for
Influenza Virus Replication." *Nature* 463 (7282):813–17.

Oguin, Thomas H., 3rd, Shalini Sharma, Amanda D. Stuart, Susu Duan, Sarah A. Scott,
Carrie K. Jones, J. Scott Daniels, Craig W. Lindsley, Paul G. Thomas, and H. Alex
Brown. 2014. "Phospholipase D Facilitates Efficient Entry of Influenza Virus, Allowing
Escape from Innate Immune Inhibition." *The Journal of Biological Chemistry* 289
(37):25405–17.

Stertz, Silke, and Megan L. Shaw. 2011. "Uncovering the Global Host Cell Requirements for
Influenza Virus Replication via RNAi Screening." *Microbes and Infection / Institut
Pasteur* 13 (5):516–25.

Tremblay, Tammy-Lynn, and Jennifer J. Hill. 2017. "Biotin-Transfer from a Trifunctional

- Crosslinker for Identification of Cell Surface Receptors of Soluble Protein Ligands.”
*Scientific Reports* 7:46574.
- Vial, Daniel, and Paula J. McKeown-Longo. 2012. “Epidermal Growth Factor (EGF)
Regulates $\alpha 5\beta 1$ Integrin Activation State in Human Cancer Cell Lines through the
p90RSK-Dependent Phosphorylation of Filamin A.” *The Journal of Biological Chemistry*
287 (48):40371–80.
- Wollscheid, Bernd, Damaris Bausch-Fluck, Christine Henderson, Robert O’Brien, Miriam
Bibel, Ralph Schiess, Ruedi Aebersold, and Julian D. Watts. 2009. “Mass-Spectrometric
Identification and Relative Quantification of N-Linked Cell Surface Glycoproteins.”
*Nature Biotechnology* 27 (4):378–86.
- Yang, J., H. Chen, I. R. Vlahov, J-X Cheng, and P. S. Low. 2007. “Characterization of the pH
of Folate Receptor-Containing Endosomes and the Rate of Hydrolysis of Internalized
Acid-Labile Folate-Drug Conjugates.” *The Journal of Pharmacology and Experimental*
*Therapeutics* 321 (2):462–68.
- Zhu, L., H. Ly, and Y. Liang. 2013. “PLC- 1 Signaling Plays a Subtype-Specific Role in
Postbinding Cell Entry of Influenza A Virus.” *Journal of Virology* 88 (1):417–24.

Reviewers' Comments:

Reviewer #1 (Remarks to the Author):

The authors have done a very thorough job of addressing the reviewer comments and the additional details added will greatly help others in the field working in this area.

Thanks.

Reviewer #2 (Remarks to the Author):

The authors thoroughly responded to the previous comments. All raised concerns have been addressed experimentally or in the discussion to full satisfaction and I recommend accepting the manuscript for publication. Clearly this study is a major advancement in the field of receptor identification.

Reviewer # 3 could not comment on this revision. We asked Reviewer #2, who has the similar expertise coverage as Reviewer #3, to comment whether (s)he thinks Reviewer #3 previous concerns have been successfully addressed. Please refer the report in the attached PDF file.

Reviewer #2 Comments on Reviewer #3's Suggestions:

To the authors:

Almost all points raised have been addressed. A short discussion on the EGF and folate receptor candidates that were found, could be added. Otherwise it seems a fine study and a valuable contribution to the receptor identification field.

Main point 1:

The discussion of new EGF receptor candidates could not be found. The authors refer the reviewer to p. 10 of the point by point response, but no discussion of the new candidates is provided there.

Similarly the discussion of new folate candidate receptors could not be found.

Regarding the discussion of known IAV receptors, this was adequately addressed by the authors.

Main point 2:

Addressed by the authors.

Main point 3:

Fully addressed by the authors.

Minor points:

All addressed and/or explained sufficiently by the authors

To the authors:

Almost all points raised have been addressed. A short discussion on the EGF and folate receptor candidates that were found, could be added. Otherwise it seems a fine study and a valuable contribution to the receptor identification field.

Main point 1:

The discussion of new EGF receptor candidates could not be found. The authors refer the reviewer to p. 10 of the point by point response, but no discussion of the new candidates is provided there. Similarly, the discussion of new folate candidate receptors could not be found.

Regarding the discussion of known IAV receptors, this was adequately addressed by the authors.

Response: On p. 10 of the point-by-point response we wrote the text below and forgot to mention our more extensive discussion on p. 16 and following pages. We copied these sections and additional relevant changes to the manuscript below.

- [...], we correctly identified EGF significantly enriched and identified five other EGF receptor candidates that have not been described before (**Supplementary Table 3**), namely monocarboxylate transporter 4 (SLC16A3), filamin-A (FLNA), peroxisomal 3-ketoacyl-CoA thiolase (ACAA1), transmembrane emp24 domain-containing protein 7 (TMED7) and sarcoplasmic/endoplasmic reticulum calcium ATPase 1 (AT2A1) (**Supplementary Table 3**). Reports of direct interactions between these proteins and EGF are not available, but it was shown before that SLC16A3 co-locates with CD147 in breast cancer cells (Gallagher et al. 2007) , which in turn is associated with EGFR in similar lipid domains (Vial and McKeown-Longo 2012) suggesting that SLC16A3 resides in the neighborhood of EGFR at the cell surface (Dai et al. 2013) . [...]
- HATRIC-LRC is a screening technology, which enables the identification of receptor candidates. In certain case scenarios, identified candidates may not be direct interaction partners of the ligand as you pointed out. Apart from the main receptor, other candidates identified could be “next door neighbors”, potentially influencing receptor activity, which were captured due to proximity to the main receptor. We are following up on this exciting possibility. Given the experimental setup, the candidates identified from HATRIC-LRC experiments can generally be the result of four reasons: (1) there is a direct interaction of the ligand with the target receptor; (2) the protein is in close proximity of the target receptor (“neighborhood protein”); (3) the protein gets upregulated in response to treatment with the ligand and gets overrepresented in the background binding of HATRIC (e. g. we use approximately 8 times more EGF than

is used for stimulation experiments) or (4) the identified candidate is a false positive. Our experiments do not allow us to delineate right away which type of interaction was observed, but the validation experiments and the cited data clearly underline the relevance of the identified proteins. The analysis pipeline was optimized to allow for identification and ranking of receptor candidates. However, the resulting data have to be analyzed carefully and more stringent receptor spaces can be defined based on the identification of positive control receptors or the ligand (e.g. EGF). Identified candidates need validation in tailor-made follow-up experiments, such as siRNA-based approaches. These approaches cannot be generalized and for every LRC application the type of follow-up experiment will depend on the type of ligand, the biological context, and the tools available for the system under study. However, we would also like to point out that the biological relevance of the neighboring proteins is not to be underestimated either. Proteins that are in close proximity of the target receptor might interfere with the activity of the actual target and are therefore relevant for future studies of the lateral cell surface interactome. HATRIC-LRC could potentially also be used to generate candidates for such studies - another exciting application of HATRIC-LRC for life science research.

Changes to the manuscript:

[...] We incubated the folate-HATRIC conjugate with 20 million HeLa Kyoto cells at pH 7.4. In the control, we added six-fold excess of unmodified folate. We detected interactions with FOLR1 and with a small set of further receptor candidates (**Fig. 2c, d; Supplementary Table 7**). None of these receptors were previously described to interact directly with folate. At the same time, we didn't identify any other known folate receptors. We speculate that other folate receptors (e. g. FOLR2) were not identified as their affinity towards folate is lower than the affinity of FOLR1 or because they are not expressed in HeLa Kyoto cells¹⁹. Related approaches studied methotrexate-based labeling of FOLR1, but didn't investigate if the compound also binds to other proteins¹⁸.

[...] Applying this filter prior to statistical analysis, we correctly identified EGF significantly enriched and identified five other EGF receptor candidates that have not been described before (Supplementary Table 3-4), namely monocarboxylate transporter 4 (SLC16A3), filamin-A (FLNA), peroxisomal 3-ketoacyl-CoA thiolase (ACAA1), transmembrane emp24 domain-containing protein 7 (TMED7) and sarcoplasmic/endoplasmic reticulum calcium ATPase 1 (AT2A1) (Supplementary Table 3-4). Reports of direct interactions between these proteins and EGF are not available, but it was shown before that SLC16A3 co-locates with CD147 in breast cancer cells (Gallagher et al. 2007) , which in turn is associated with EGFR in similar lipid

domains (Vial and McKeown-Longo 2012) suggesting that SLC16A3 resides in the neighbourhood of EGFR at the cell surface (Dai et al. 2013).

Discussion for both sections added to the manuscript:

[...] Given the experimental setup, the candidates identified from HATRIC-LRC experiments can generally be the result of four scenarios (1) there is a direct interaction of the ligand with the target receptor; (2) the protein is in close proximity of the target receptor ("neighbourhood protein"); (3) the protein gets upregulated in response to treatment with the ligand and gets overrepresented in the background binding of HATRIC (e. g. we use approximately 8 times more EGF than is used for stimulation experiments) or (4) the identified candidate is a false positive. A single HATRIC-LRC experiment does not allow us to delineate which type of interaction was observed, but the validation experiments and the cited data clearly underline the biological relevance of the identified proteins. The analysis pipeline was optimized to allow for the identification and ranking of receptor candidates. However, the resulting data have to be analyzed carefully and more stringent receptor spaces can be defined based on the identification of positive control receptors or the ligand (e.g. EGF). Identified candidates need validation in tailor-made follow-up experiments, such as siRNA-based approaches. [...]

Main point 2:

Addressed by the authors.

Main point 3:

Fully addressed by the authors.

Minor points:

All addressed and/or explained sufficiently by the authors